# A protein adaptor mediating Ap4A-dependent control of protein acetylation

Liujuan Zheng [1,2], Megan K. M. Young[3], Wieland Steinchen[2], Zhiyong Guo [4], Ekaterina Jalomo-Khayrova[2], Bobby Xuanyu Liu[3], Fabiana Burchert [2], Patricia Bedrunka [2], Christopher-Nils Mais[2], Jan Pané-Farré[2], Mathias Girbig [1], Uwe Linne [2], Aude Trinquier [3], Aitao Li [4], Georg Hochberg[1], Johannes Freitag [2], Jue D. Wang [3] ✉ & Gert Bange [1,2] ✉

Reversible lysine acetylation is a highly conserved post-translational modification across all domains of life controlling diverse cellular processes such as metabolism and gene expression. However, the regulation of protein acetylation remains poorly understood. Here, we report a regulatory system in *Bacillus subtilis* that controls the activity of the histone deacetylase (HDAC)-like protein AcuC, which has multiple substrates including acetyl-CoA synthetase and translation elongation factor. We show that AcuC is inhibited via formation of a stable complex with the hitherto uncharacterized protein AcuB. We furthermore demonstrate that the alarmone diadenosine tetraphosphate (Ap4A) binds to the cystathionine beta-synthase (CBS) domain of AcuB, thereby stabilizing AcuB and further enhancing the inhibition of AcuC. In summary, this study identifies AcuB as an Ap4A regulated deacetylation inhibitor, revealing a uncharacterized molecular mechanism to control HDAC-like proteins. Thus, the alarmone Ap4A modulates protein (de)acetylation, pointing towards a regulatory network that connects stress response, protein acetylation, and acetyl-CoA biosynthesis.

Reversible lysine acetylation is an evolutionarily conserved post-translational modification, which plays a crucial role in a variety of cellular processes, including the regulation of metabolism and gene expression[1–4]. Lysine acetylation is catalyzed by different acetyltransferases, e.g., Gcn5-related N-acetyltransferases (GNATs), the p300/CBP family, and the MYST family, using acetyl-CoA (Ac-CoA) as acetyl donor. Lysine acetylation can also occur in the absence of a dedicated enzyme via acetyl-phosphate (AcP) or Ac-CoA[3–5]. Deacetylation is mediated by either of two classes of lysine deacetylases – (i) NAD-dependent sirtuins and (ii) $Zn^{2+}$-dependent histone deacetylases (HDACs) and HDAC-like proteins[6–9].

Advances in proteomics and mass spectrometry have uncovered pervasive lysine acetylation in diverse bacterial proteins[2,10]. However,

how protein acetylation is regulated is poorly understood. One example is the reversible acetylation of the catalytic lysine in acetyl-CoA synthetases (Acs), which inactivates the enzyme[11–13]. In *Bacillus subtilis*, Acs (AcsA) acetylation is catalyzed by the GNAT AcuA, while deacetylation (and thus reactivation) is mediated by the $Zn^{2+}$-dependent HDAC like protein AcuC[13] (Fig. 1a). AcuA and AcuC are encoded in an operon that also contains *acuB*, a gene of unknown function[13–16].

Nucleotide second messengers such as cyclic AMP (cAMP), guanosine-(penta)tetraphosphate ((p)ppGpp), and dinucleoside polyphosphates are crucial for stress signaling across bacteria and archaea[17–21]. Dinucleoside polyphosphates occur in all domains of life (e.g., as a side product of aminoacyl-tRNA synthetases) and have been proposed as evolutionary conserved alarmones[22,23]. In particular, diadenosine

[1]Max Planck Institute for Terrestrial Microbiology, Marburg, Germany. [2]Marburg University, Center for Synthetic Microbiology (SYNMIKRO) & Department of Chemistry, Marburg, Germany. [3]Department of Bacteriology, University of Wisconsin-Madison, Madison, WI, USA. [4]State Key Laboratory of Biocatalysis and Enzyme Engineering, Hubei Key Laboratory of Industrial Biotechnology, School of Life Sciences, Hubei University, Wuhan, China. ✉e-mail: wang@bact.wisc.edu; gert.bange@synmikro.uni-marburg.de

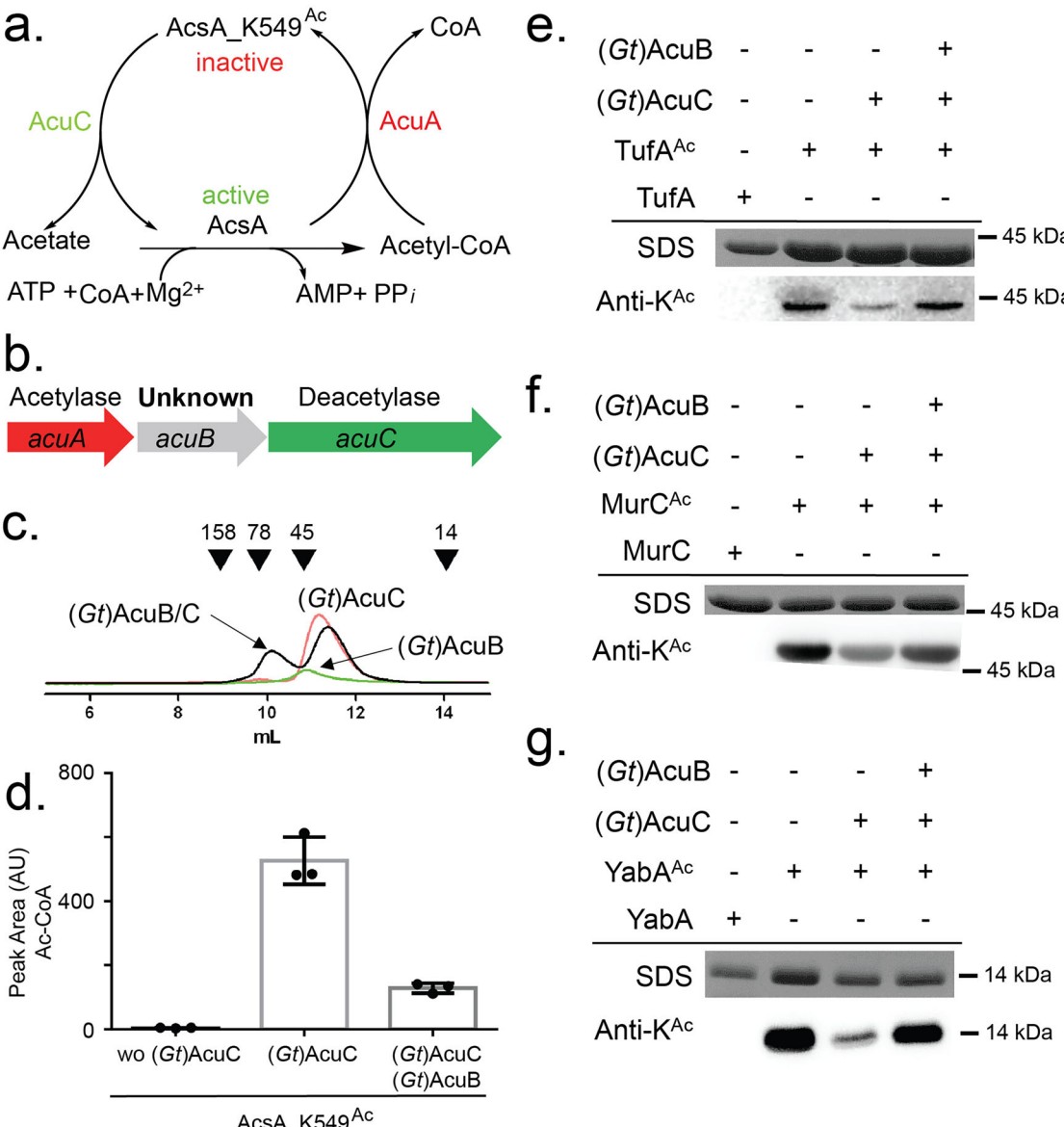

**Fig. 1 | AcuB inhibits the deacetylase activity of AcuC. a** Regulation of acetyl-CoA production from acetate via acetylation of the catalytic lysine. Acetyl-CoA synthetase AscA is inactivated by AcuA and activated by AcuC in *B. subtilis*. **b** The *acuABC* operon of *B. subtilis*: *acuA* encodes a GNAT enzyme responsible for acetylation of AcsA at lysine 549 (AcsA_K549), *acuB* encodes a protein of unknown function and *acuC* encodes an HDAC-like protein that deacetylates AcsA_K549$^{Ac}$.
**c** Chromatograms from analytical size-exclusion chromatography showing the profiles of (*Gt*)AcuB (green), (*Gt*)AcuC (orange), and the (*Gt*)AcuB-(*Gt*)AcuC complex (black). Numbers above denote elution volumes of standard proteins with the

indicate molecular masses (in kDa). **d** Results of an AcsA activity assay starting with acetylated (inactive) enzyme AcsA_K549$^{Ac}$. Either (*Gt*)AcuC or (*Gt*)AcuC-(*Gt*)AcuB together with 20 μM Zn$^{2+}$ were added to the reaction mixture. Plotted is the mean of three experiments. Error bars denote standard error of the mean. **e**–**g** Deacetylation activity of (*Gt*)AcuC on various targets in the absence or presence of AcuB. TufA$^{Ac}$ (**e**), MurC$^{Ac}$ (**f**) and YabA$^{Ac}$ (**g**). The upper panels show the Coomassie-stained gels, while the lower panels display the immunoblots probed with an acetylation-specific antibody. Source data underlying (**c**–**g**) are provided as a Source Data file.

tetraphosphate (Ap4A) accumulates under various stress conditions in diverse bacteria such as *Salmonella typhimurium*, *Escherichia coli*, and *Bacillus subtilis*[24–27]. Previously, we demonstrated that Ap4A regulates GTP biosynthesis by binding to the CBS (Cystathionine-Beta-Synthase) domain of inosine-5′-monophosphate dehydrogenase (IMPDH) in *B. subtilis*[27]- a mechanism also found in the fungi[28]. Recent proteomic studies in human embryonic kidney cell lysates revealed nearly 100 potential Ap4A-binding proteins[29], suggesting its role as an important signaling molecule also in the eukaryotes. However, only little mechanistic knowledge about Ap4A signaling exits.

In this work, we present that similar to IMPDH the *B. subtilis* protein of unknown function AcuB contains two CBS domains, which

led us to hypothesize that Ap4A might interact directly with AcuB. Thus, we set out to explore the molecular function of AcuB. We demonstrate that it directly interacts with the HDAC-like deacetylase AcuC, thereby inhibiting deacetylation of various proteins, including AcsA, the elongation factor EF-Tu, the cell wall biosynthesis enzyme MurC, and the DNA replication regulator YabA. DRaCALA (direct radial capillary action of ligand assays) screening revealed that AcuB indeed binds Ap4A suggesting a potential connection between Ap4A and the bacterial acetylation–deacetylation machinery. The crystal structure of AcuB bound to Ap4A reveals that AcuB comprises three domains: two CBS domains and one ACT (aspartate kinase-chorismate mutase-tyrA) domain. The ACT domain mediates interaction with AcuC and blocks

its active site. The CBS domains bind Ap4A with high affinity, increasing its thermal stability and binding to AcuC thereby accelerating the inhibitory effect of AcuB on AcsA dependent Ac-CoA production. Together, our findings introduce a regulatory mechanism that connects alarmone signaling to protein acetylation and biosynthesis of Ac-CoA.

## Results

### AcuB interacts with AcuC

The enzyme activity of *B. subtilis* AcsA is tightly regulated through reversible acetylation of its catalytic lysine residue (K549), mediated by the opposing actions of the acetyltransferase AcuA and the deacetylase AcuC (Fig. 1a). AcuA and AcuC reside within an operon together with AcuB, a protein of unknown function (Fig. 1b). We and others have shown that AcsA can form a stable complex with AcuA but not with AcuB[16,30].

To identify interaction partners of AcuB, we performed in vivo pull-down assays in *B. subtilis* using Streptavidin II-tagged AcuB expressed from the replicative plasmid pLike_Rep35. The empty plasmid pLike_Rep expressing only the tag served as a negative control. Comparative mass spectrometry analysis of eluates identified AcuC as one of the most abundant proteins specifically enriched in the AcuB samples (Supplementary Fig. 1, Supplementary datasets 1 and 2) suggesting a physical interaction between AcuB and AcuC in *B. subtilis*.

To confirm the interaction of AcuB and AcuC, we recombinantly produced both *B. subtilis* proteins in *Escherichia coli* BL21(DE3) and purified both proteins via a two-step protocol consisting of a nickel-affinity chromatography followed by a size-exclusion chromatography (SEC). During purification, *B. subtilis* AcuB aggregated severely. As proteins from thermophilic organisms often display improved biochemical properties and stability[31], we instead used the AcuB homolog from *Geobacillus thermodenitrificans* NG80-2 (*Gt*) (Supplementary Fig. 2). Indeed, (*Gt*)AcuB did not aggregate allowing the purification of soluble (*Gt*)AcuB with a high yield. For consistency reasons, we purified the (*Gt*)AcuC homolog as well (Supplementary Fig. 3). To validate complex formation, we performed an analytical SEC experiment, which demonstrated the formation of a stable heteromeric (*Gt*)AcuB/C complex (Fig. 1c, Supplementary Fig. 4).

### (*Gt*)AcuB inhibits the deacetylase activity of (*Gt*)AcuC towards multiple targets including AcsA

Next, we assessed the influence of (*Gt*)AcuB binding on (*Gt*)AcuC deacetylase activity. It was known that (*Gt*)AcuC removes the acetyl group from the catalytic lysine 549 of AcsA, thereby reactivating AcsA to synthesize Ac-CoA[13]. To examine the effect of (*Gt*)AcuB on the activity of (*Gt*)AcuC, we prepared acetylated AcsA (AcsA_K549$^{Ac}$) and confirmed that it lost its ability to synthesize Ac-CoA as previously described[30] (Fig. 1d). As expected, incubation of (*Gt*)AcuC with AcsA_K549$^{Ac}$ in the presence of $Zn^{2+}$ restored the production of Ac-CoA, confirming that AcuC deacetylates AcsA_K549$^{Ac}$ to reactivate the enzyme. However, when (*Gt*)AcuB was premixed with (*Gt*)AcuC and then incubated with AcsA_K549$^{Ac}$, Ac-CoA production decreased by approximately 75% demonstrating that (*Gt*)AcuB inhibits the activity of (*Gt*)AcuC (Fig. 1d).

A previous study showed that in *B. subtilis*, the translation elongation factor TufA (EF-Tu) is acetylated by a yet unknown acetyltransferase or nonenzymatical manner and subsequently deacetylated by (*Bs*)AcuC[32]. To investigate the impact of (*Gt*)AcuB on the deacetylation activity of (*Gt*)AcuC towards TufA, we overexpressed (*Bs*)TufA in *E. coli* and purified the His-tagged fusion protein via nickel affinity chromatography and gel filtration. Immunoblot analysis revealed that TufA was not acetylated under these conditions (Fig. 1e). Since high Ac-CoA concentrations are sufficient to acetylate proteins at least in vitro[32], we incubated (*Bs*)TufA with 1 mM Ac-CoA overnight, yielding acetylated elongation factor (*Bs*)TufA$^{Ac}$. The acetylation was confirmed

by a strong anti-acetyl-lysine antibody signal (Fig. 1e). Incubation with (*Gt*)AcuC strongly reduced this signal, indicating (*Gt*)AcuC was able to remove the acetylation from (*Bs*)TufA. Importantly, pre-incubation of (*Gt*)AcuC with (*Gt*)AcuB prior to adding (*Bs*)TufA$^{Ac}$ resulted in a much stronger signal, showing that AcuB also inhibits deacetylation activity of (*Gt*)AcuC on the elongation factor TufA (Fig. 1e).

Another putative target of AcuC is MurC, a protein involved in cell wall biosynthesis[33]. Previous work showed that depletion of the acetyltransferase AcuC reduced MurC acetylation. We successfully generated acetylated MurC (MurC$^{Ac}$) and confirmed that it is a substrate of AcuC (Fig. 1f). Again, addition of AcuB inhibited the deacetylation of MurC$^{Ac}$ (Fig. 1f). A Yeast two-hybrid assay performed previously suggested an interaction of AcuB and the DNA replication regulator YabA[34]. Although we could not establish an AcuB/YabA interaction with the purified proteins (Supplementary Fig. 5), we observed that acetylated (*Bs*)YabA (YabA$^{Ac}$) is deacetylated by (*Gt*)AcuC (Fig. 1g), and again (*Gt*)AcuB inhibited the AcuC-dependent deacetylation of (*Bs*)YabA (Fig. 1g). In summary, our experiments demonstrate that AcuC can efficiently remove acetyl groups from various proteins and this activity is regulated by AcuB.

### The ACT-domain of AcuB interacts with AcuC

Our attempts to obtain a high-resolution structure of the AcuB-AcuC complex using X-ray crystallography or cryo-electron microscopy were unsuccessful. AlphaFold[35] analysis revealed that AcuB consists of two CBS domains at the N-terminus and one ACT domain at the C-terminus (Fig. 2a). AcuC essentially consists of one domain resembling the fold of a typical histone deacetylase (HDAC) class I domain (HDAC_AcuC_-like domain)[36] (Fig. 2a). To identify the interaction interface between the two proteins, we applied hydrogen-deuterium exchange mass spectrometry (HDX-MS). The HDX-MS results uncovered the interaction interface between AcuB and AcuC. In (*Gt*)AcuB, HDX protection was observed in the ACT domain (residues R184–Y202) and the linker region between the ACT and CBS domains (residues Q129–I140), indicating that they are part of the complex interface (Supplementary Fig. 6). For (*Gt*)AcuC, significant HDX protection was detected in multiple regions, including residues A84–H100, S119–A152, V167–F181, S191–L233, A260–Y276, A297–W306, and A368–Y379. These regions correspond to areas surrounding the active site cavity (Fig. 2b, Supplementary Fig. 7), as seen in the structure PDB ID: 1ZZ1[37] of the homolog from *Alcaligenaceae bacterium*, in which the inhibitor suberoylanilide hydroxamic acid (SAHA) was bound in the active center. Our HDX-MS data are corroborated by the AlphaFold model, which also predicts that the ACT domain of AcuB interacts with the regions surrounding the active site of AcuC (Fig. 2b, Supplementary Fig. 8).

More detailed analysis revealed that the interface between (*Gt*)AcuB and (*Gt*)AcuC is stabilized by multiple interactions, including salt bridges between (*Gt*)AcuB_E4–(*Gt*)AcuC_R362, (*Gt*)AcuB_D89–(*Gt*)AcuC_K378, and (*Gt*)AcuB_E92–(*Gt*)AcuC_R197, as well as π–π and hydrophobic interactions between (*Gt*)AcuB_W206–(*Gt*)AcuC_Y197 and (*Gt*)AcuB_M188–(*Gt*)AcuC_F200/F143 (Supplementary Fig. 9A). To validate these predicted interfacial contacts, we constructed single-point mutants (*Gt*)AcuB_D89K, E92R, M188A and (*Gt*)AcuC_R197E, Y198A, R362E, and performed pulldown assays (Supplementary Fig. 9B). The mutations D89K, E92R, and M188A in (*Gt*)AcuB reduced complex formation, while R197E and Y198A of (*Gt*)AcuC moderately weakened the interaction. In contrast, (*Gt*)AcuC_R362E showed minimal effect. These results confirm the key interfacial residues mediating the (*Gt*)AcuB–AcuC interaction.

In addition, we conducted an in vitro pull-down assay. We fused the (*Gt*)AcuB_ACT domain to a GST-tag, overexpressed this fusion protein in *E. coli*, and used (*Gt*)AcuC as prey. A band corresponding to (*Gt*)AcuC (approximately 45 kDa) was observed when incubated with (*Gt*)AcuB_ACT, confirming a direct interaction (Fig. 2c). Hence, the ACT domain of AcuB contains the binding site for AcuC.

 

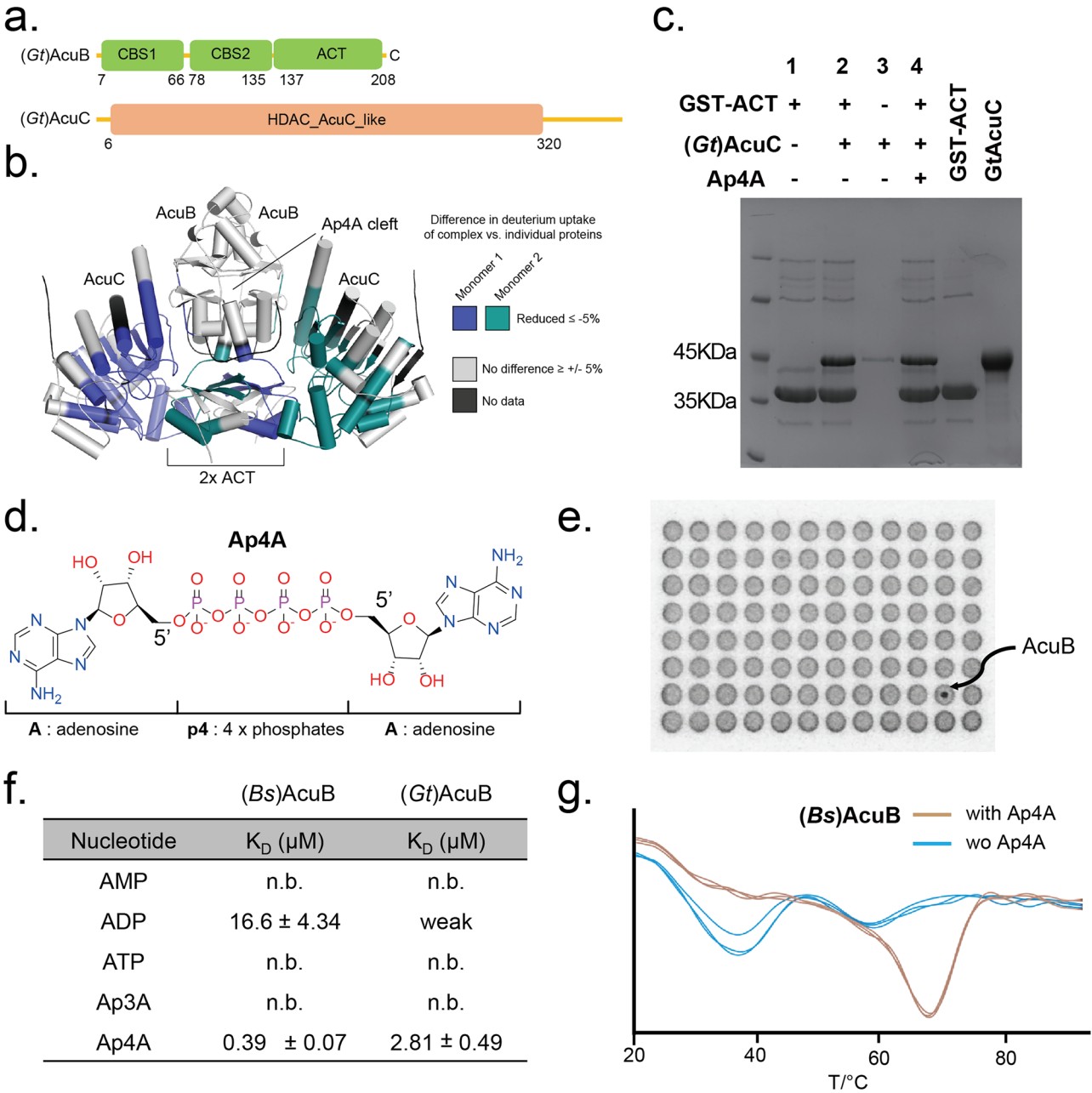

**Fig. 2 | AcuB interacts with di-adenosine-tetraphosphate (Ap4A). a** Schematic representation of the domain architectures of (*Gt*)AcuB and (*Gt*)AcuC, drawn to scale. CBS (green) Cystathionine-Beta-Synthase domain, ACT Aspartate kinase-chorismate mutase-tyrA domain, HDAC – Histone deacetylase. Numbers indicate amino acid positions. **b** HDX-analysis showing differences in the (*Gt*)AcuB-(*Gt*)AcuC complex (model produced by AlphaFold 3)[35], compared to (*Gt*)AcuB and (*Gt*)AcuC individually. **c** Coomassie-stained SDS-PAGE of a GST-pulldown assay using a GST-tagged (*Gt*)AcuB-ACT domain as bait to assess its interaction with (*Gt*)AcuC.

**d** Chemical structure of Ap4A (P1,P4-Bis(5′-adenosyl) tetraphosphate). **e** Results of a Differential Radial Capillary Action of Ligand Assay (DRaCALA) for identifying Ap4A binding partners in *Bacillus subtilis*. **f** Dissociation constants ($K_D$s), determined by ITC analysis, for the binding of different nucleotides (i.e., AMP, ADP, ATP, Ap3A and Ap4A) to (*Bs*)AcuB and (*Gt*)AcuB. n.b. no binding. **g** Thermostability assays of (*Bs*)AcuB: (*Bs*)AcuB alone (blue) and (*Bs*)AcuB with 500 μM Ap4A (brown) demonstrating the stabilizing effect of Ap4A on (*Bs*)AcuB. Source data underlying (**c**, **f**, **g**) are provided as a Source Data file.

## AcuB binds to the alarmone Ap4A

Our finding that the ACT domain of AcuB is the primary binding site of AcuC suggests that the N-terminal CBS domains of AcuB fulfill another function. CBS domains are evolutionarily conserved and occur in a wide range of species from bacteria to humans. Typically they come in pairs and bind adenosine-containing molecules such as AMP, ATP[38] or Ap4A (Fig. 2d)[27]. Using tagged ORF libraries comprising >5000 *B. anthracis* genes overexpressed in *E. coli*[39], we performed a DRaCALA[39] with in vitro synthesized [32]P-Ap4A to identify potential binding targets. Notably, AcuB appeared as a binder of Ap4A in *Bacillus anthracis* just

as IMPDH, which we have confirmed previously[27] (Fig. 2e). To test this screening result more directly, we performed isothermal calorimetry (ITC) experiments with different adenosine-containing nucleotides, such as AMP, ADP, ATP, Ap3A and Ap4A. Purified (*Bs*)AcuB protein showed moderate binding affinity for Ap4A, with a dissociation constant ($K_D$) value of $0.39 \pm 0.07$ μM (Fig. 2f). (*Gt*)AcuB exhibits a slightly weaker binding affinity for Ap4A, with a $K_D$ value of $2.81 \pm 0.49$ μM. The other nucleotides that were tested in ITC, showed no binding or only weak affinity to (*Gt*)AcuB (Fig. 2f, Supplementary Fig. 10). Thus, AcuB binds to Ap4A probably via its CBS domain-containing N-terminus.

During the purification of (*Bs*)AcuB, the protein showed slower precipitation if Ap4A was added compared to our original attempts (see above). This suggests that Ap4A stabilizes the protein and prevents its denaturation. To quantify this effect, we assessed the thermal stability of (*Bs*)AcuB using Nano Differential Scanning Fluorimetry (nanoDSF). Remarkably, the presence of Ap4A raised the melting point of (*Bs*)AcuB by approximately 30 °C, from 38.0 °C to 68.0 °C (Fig. 2g), compared to (*Bs*)AcuB alone. Next, we investigated the stability of the variant (*Bs*)AcuB_R33E (which lost the binding affinity to Ap4A, see below). This variant was slightly more thermostable than the wild-type protein. Furthermore, the addition of Ap4A did not result in a similar increase in thermostability (38.3 °C–43.8 °C; Supplementary Fig. 11A). These results suggest that Ap4A binding substantially stabilizes AcuB under conditions where Ap4A levels are elevated, such as during heat stress. Supporting this notion, we observed that (*Gt*)AcuB, derived from the moderately thermophilic bacterium *Geobacillus thermodenitrificans*, showed only a modest increase in thermostability upon Ap4A addition−approximately 5 °C (from 63.1 °C−68.4 °C). (Supplementary Fig. 11B). It is worth mentioning that ADP also shows a μM binding affinity to (*Bs*)AcuB. To further investigate the binding specificity of (*Bs*)AcuB to Ap4A, we performed competition experiments using a thermoshift assay. In these experiments, AcuB was incubated with both ADP and Ap4A, with ADP concentrations set 5 times higher than Ap4A. Despite the excess ADP, the thermal shift profile was nearly identical to that observed when Ap4A was present alone, confirming that Ap4A preferentially binds to (*Bs*)AcuB also in competitive binding conditions. As ADP binds to (*Bs*)AcuB and changes its thermal stability – albeit to a lesser extent – it is possible that variations of ADP levels regulate AcuB stability in non-stressed cells, a phenomenon which merits future investigation. Together our data show that AcuB is an Ap4A stabilized AcuC inhibitor.

## Structural basis of Ap4A-binding to AcuB

To elucidate the binding mode of Ap4A to AcuB, we attempted to crystallize both (*Gt*)AcuB and (*Bs*)AcuB in the presence of Ap4A. Crystals of (*Bs*)AcuB were obtained in the presence of Ap4A. The crystal structure of (*Bs*)AcuB in complex with Ap4A was determined at a resolution of 2.4 Å (PDB ID: 9QS7, Supplementary Table 1). The overall structure of (*Bs*)AcuB consists of two CBS domains in the N-terminus and one ACT domain in the C-terminus (Fig. 3a). (*Bs*)AcuB forms a homodimer, with the two chains cooperating in a configuration similar to that of its homolog TTHA0829 from *Thermus thermophilus* (PDB ID: 5AWE)[40] (Fig. 3b, Supplementary Fig. 12). This suggests that TTHA0829 also represents an AcuB protein. During data processing of the electron density map for AcuB, two additional electron densities were detected between its four CBS domains, which could be identified as Ap4A molecules (Supplementary Fig. 13). They bind into extended surface grooves present on the opposite sides of the CBS-domain part of the AcuB homodimer (Fig. 3b). Given the high structural alignment between (*Bs*)AcuB and the (*Gt*)AcuB AlphaFold model (Supplementary Fig. 14), we infer that two Ap4A molecules also bind within the CBS domains of (*Gt*)AcuB. The results from HDX-MS experiments align well with the X-ray structure, confirming that Ap4A binds to and stabilizes the CBS domain (see below, Supplementary Fig. 6).

Further inspection using LigPlot[41,42] reveals the highly symmetrical coordination of the two Ap4A molecules and provides a molecular model explaining how they might stabilize the AcuB homodimer (Fig. 3b, c). Each Ap4A molecule binds to both units of the homodimer in a crisscross fashion. One adenosine moiety forms hydrogen bonds with the backbone carbonyl groups of His34 and Ile12 from one monomer, while the other adenosine moiety binds to the backbone carbonyls of Cys105 and Val183 from the other monomer. These interactions are reciprocal in the symmetric dimer, with the second Ap4A molecule binding in the opposite orientation to the pair of

monomers. In addition, the two ribose moieties of each Ap4A molecule interact with Asp122 through van der Waals forces (Fig. 3c). Moreover, the positively charged side chains of Arg33, Arg51, Lys120, His34, along with the backbone of Gly104 from both monomers form hydrogen bonds with the phosphate moieties of the two Ap4A molecules (Fig. 3b, c). Notably, Arg33 and His34 from both monomers bridge the two Ap4A molecules via hydrogen bonds (Fig. 3d). These interactions indicate a tight binding mode where Ap4A stabilizes and induces significant conformational changes in the (*Bs*)AcuB structure.

Next, variations of Arg33, Arg51 into glutamate, and Asp122 into alanine were introduced in (*Bs*)AcuB to assess their impact on Ap4A binding via ITC. The R33E variant abolished binding, highlighting its critical role (Fig. 3f, Supplementary Fig. 10). The R51E and D122A variants showed reduced binding affinities with $K_D$s of 28.6 ± 15.0 μM and 0.72 ± 0.08 μM, respectively, compared to the wildtype protein (Supplementary Figs. 10 and 15). Homology analysis of 200 AcuB sequences revealed that R33 and His34 are highly conserved residues, implying that AcuB orthologs are conserved Ap4A binders (Fig. 3e)[43]. Together, our data show that the CBS domains of the AcuB homodimer coordinate two Ap4A molecules in a symmetric fashion.

## Ap4A binding induces conformational changes in AcuB and enhances its inhibitory function

To examine how Ap4A binding affects AcuB structure and function, we analyzed the effects of Ap4A on (*Gt*)AcuB using HDX-MS (Supplementary Table 2 and Supplementary Dataset 3). This method revealed significant Ap4A-induced changes in the CBS domain of AcuB. Residues V3–L14, V48–F61, D81–F90, and G116–T121 exhibited reduced deuterium uptake, indicating stabilization of these regions upon Ap4A binding. In contrast, residues D122–L127, encompassing a helix and the loop connecting the CBS and ACT domains, showed increased deuterium uptake, suggesting enhanced flexibility in this area (Fig. 4a, Supplementary Fig. 6). Thus, Ap4A binding induces significant conformational changes in the CBS domain of AcuB and increases the flexibility of the linker. The dynamic structural rearrangements of AcuB upon Ap4A binding suggest that Ap4A may impact the inhibitory action of AcuB on the AcuC deacetylase. However, we did not observe any HDX changes in the (*Gt*)AcuB-AcuC complex with or without Ap4A, suggesting that Ap4A does not directly induce large conformational changes (Supplementary Figs. 6 and 7).

To assess the precise role of Ap4A binding, we investigated the effect of Ap4A on both the AcuB/AcuC interaction and the inhibitory role of AcuB on AcuC. To validate the interaction between AcuB and AcuC, we performed pulldown assays using strep-tagged (*Gt*)AcuB as bait and (*Gt*)AcuC as prey. A band corresponding to (*Gt*)AcuC was only observed in the presence of (*Gt*)AcuB, confirming the formation of a (*Gt*)AcuB-AcuC complex (Fig. 4b). Notably, the presence of Ap4A increased the intensity of the (*Gt*)AcuC band compared to (*Gt*)AcuC alone, whereas other nucleotides, such as ATP, ADP, and AMP did not show this increase (Fig. 4b). Interestingly, when Ap4A was added to the (*Gt*)AcuB_ACT/(*Gt*)AcuC mixture, no increase in (*Gt*)AcuC band intensity was observed (Fig. 2c). This finding indicates that while the (*Gt*)AcuB_ACT domain can interact with (*Gt*)AcuC (see also above), Ap4A does not affect this interaction, likely due to its specific binding to the CBS domain of AcuB (Figs. 3b, 4a). In addition, we employed mass photometry (MP) to better assess the effect of Ap4A on the stability of the (*Gt*)AcuB-AcuC complex. When 500 nM (*Gt*)AcuB, 500 nM (*Gt*)AcuC, and 1 mM Ap4A were premixed, resulting in final protein concentrations of 25 nM and a final Ap4A concentration of 50 μM for MP analysis, the intensity of the complex (100 kDa peak) increased approximately 2.5-fold (Fig. 4c). No increase was observed upon the addition of AMP, ADP, or ATP (Fig. 4c, Supplementary Fig. 16), which shows a similar stability to the (*Gt*)AcuB−AcuC complex in the absence of any nucleotide (approximately 10% of the total protein forms a complex in this assay). Such enhancement of complex formation was

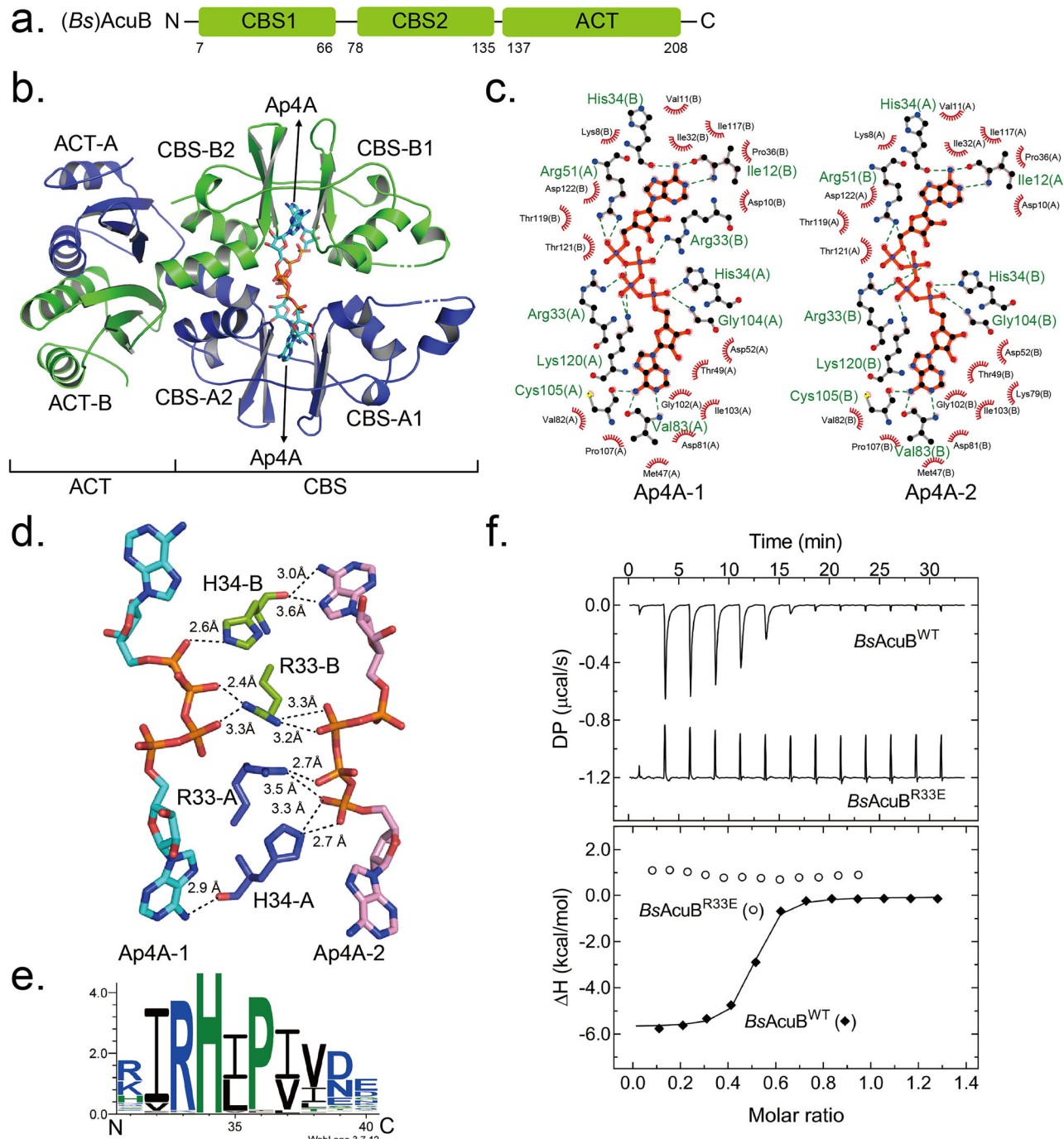

**Fig. 3 | Crystal structure of (Bs)AcuB bound to Ap4A. a** Schematic representation of the domain architecture of (Bs)AcuB. **b** Cartoon depiction of the crystal structure of the (Bs)AcuB-Ap4A complex (PDB ID: 9QS7). **c** Binding analysis of the coordination of two Ap4A molecules within (Bs)AcuB was performed using LigPlot[41,42]. **d** Arg33 and His34 residues bridging the two Ap4A molecules. **e** WebLogo[43] analysis of AcuB homologues derived from a sequence alignment of 200 AcuB homologues, highlighting Arg33 and His34 as conserved residues. **f** ITC analysis showing that the (Bs)AcuB_R33E mutant does not bind to Ap4A.

not observed in the (Gt)AcuB_R33E mutant upon Ap4A addition (Fig. 4c). These results confirm that (Gt)AcuB forms a stable complex with (Gt)AcuC, and that this interaction is enhanced by Ap4A.

Next, we examined if Ap4A binding has an impact on the function of AcuB as AcuC inhibitor. (Gt)AcuC efficiently deacetylates AcsA_K549[Ac], resulting in an increase in Ac-CoA production over time (Figs. 4d, 1d). However, when (Gt)AcuB and (Gt)AcuC were premixed, Ac-CoA production was notably reduced compared to (Gt)AcuC alone. Adding Ap4A to the (Gt)AcuB and (Gt)AcuC mixture further suppressed Ac-CoA production (Fig. 4d). In contrast, (Gt)AcuB_R33E, which cannot bind Ap4A anymore (Fig. 3f), did not inhibit Ac-CoA

production beyond the effect of (Gt)AcuB_WT, even in the presence of Ap4A (Fig. 4d) demonstrating that Ap4A inhibits deacetylation activity of (Gt)AcuC on AcsA_K549[Ac] through its interaction with (Gt)AcuB.

## Mechanistic insight into mediated AcuB inhibition and the effect of Ap4A

Bioinformatic analysis revealed that AcuC is a class I HDAC homolog – it shares conserved catalytic residues and catalyzes deacetylation in a very similar way (Supplementary Figs. 17 and 18)[2,44]. The predicted AlphaFold model of (Gt)AcuC resembles the structures of *Homo sapiens* HDAC ((Hs)HDACs) 1–3 and 8[45–48], respectively. Key residues

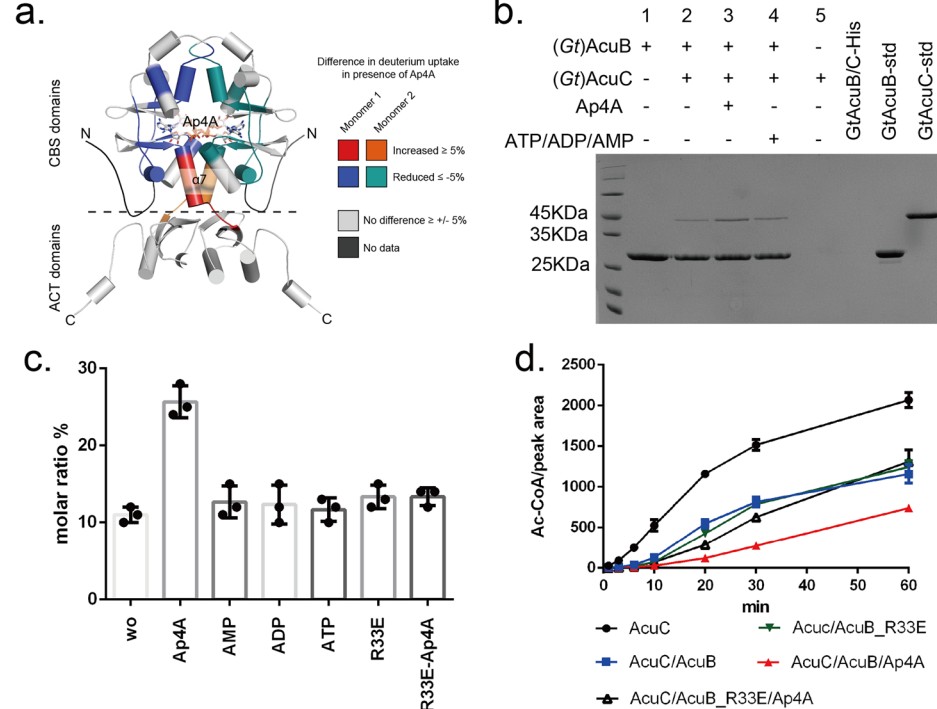

**Fig. 4 | Ap4A modulates the AcuB-dependent inhibition of AcuC. a** HDX analysis showing differences in the (*Gt*)AcuB-Ap4A complex compared to (*Gt*)AcuB alone. (*Gt*)AcuB model was produced by AlphaFold 3. **b** Coomassie-stained SDS-PAGE of a Strep-pulldown assay using Strep-(*Gt*)AcuB as bait to assess its interaction with (*Gt*)AcuC in the presence of different nucleotides. **c** Mass photometry analysis of the (*Gt*)AcuB-AcuC complex in the presence of various nucleotides. A mixture of 500 nM (*Gt*)AcuB and (*Gt*)AcuC was incubated with 1 mM of Ap4A, AMP, ADP, ATP, and without nucleotide. In the final assay these stocks are approximately 20-fold diluted resulting in concentrations of 25 nM protein and 50 μM nucleotide for measurement. The masses observed for (*Gt*)AcuB, (*Gt*)AcuC, and the (*Gt*)AcuB-AcuC complex were 54 kDa, 48 kDa, and 100 kDa, respectively. R33E means use (*Gt*)AcuB_R33E replacing (*Gt*)AcuB to form complex with (*Gt*)AcuC. The y axis molar ratio represents the N counts (peak of (*Gt*)AcuB-AcuC complex)/N counts (total) in percent. Plotted are the means of three independent experiments. Error bars represent standard deviation of the mean. **d** Deacetylation activity of (*Gt*)AcuC on AcsA_K549$^{Ac}$, showing inhibition by (*Gt*)AcuB, with Ap4A further enhancing the inhibition. Data points: AcuC on AcsA_K549$^{Ac}$ (black circles), AcuB plus AcuC on AcsA_K549$^{Ac}$ (blue squares), AcuB, AcuC plus Ap4A on AcsA_K549$^{Ac}$ (red triangles), AcuB_R33E plus AcuC on AcsA_K549$^{Ac}$ (green triangles), AcuB_R33E, AcuC plus Ap4A on AcsA_K549$^{Ac}$ (black triangles). The peak area (Ac-CoA) is produced in the presence of AcsA_K549$^{Ac}$. Data are presented as mean values ± SD and the error bars represent the standard deviation from three independent experiments. Source data underlying (**b**– **d**) are provided as a Source Data file.

such as D259, D170, and H172 coordinate the Zn$^{2+}$ ion crucial for binding to the acetyl moiety of acetylated lysine. Additionally, the proton donor/acceptor residues Y303, H133, and H134 are conserved in both AcuC and the (*Hs*)HDACs, suggesting a similar deacetylation mechanism (Fig. 5a, Supplementary Fig. 17). Notably, the active center residues exhibited protection in the HDX experiment investigating the (*Gt*)AcuB-(*Gt*)AcuC complex indicating that (*Gt*)AcuB affects key residues in the active center of (*Gt*)AcuC (Supplementary Figs. 6 and 7). Interestingly, the HDAC inhibitor SAHA was previously co-crystallized with the (*Hs*)HDAC8[48] (Fig. 5b). In the superimposed structures of the (*Gt*)AcuC–AcuB complex and the (*Hs*)HDAC8–SAHA complex, AcuB_M188 aligns closely with the SAHA binding site, suggesting that AcuB may occlude the substrate entry/exit tunnel of AcuC (Fig. 5c). This interaction was supported by the HDX experiment, which revealed protection in the loop containing AcuB_M188 and the active sites of AcuC in the HDX analysis (Fig. 5f).

To investigate how Ap4A enhances the inhibitory activity of AcuB on AcuC, we conducted molecular dynamics (MD) simulations. We used the AlphaFold model as the starting structure, as it showed high confidence in the predicted complex with high Per-residue predicted Local Distance Difference for all chains (pLDDT>85, Supplementary Fig. 19). Initially, a 200 nano seconds (ns) accelerated MD simulation was performed at 300 K on the (*Gt*)AcuB dimer. The results indicate that when two Ap4A molecules bind to the active site, the RMSF (Root Mean Square Fluctuation) values are significantly lower compared to the *apo* state (Fig. 5d), suggesting that ligand binding reduces

conformational flexibility and stabilizes the protein. This finding aligns well with our data obtained in the thermostability assay (Fig. 2g). The increase in stability may facilitate the binding of the (*Gt*)AcuB dimer to (*Gt*)AcuC. To further explore this effect, we increased the simulation temperature to 450 K and conducted MD simulations on the (*Gt*)AcuB-AcuC complex to compare the two states. It should be noted that the simulations at 450 K were not intended to mimic experimental conditions but rather to accelerate conformational sampling and enable comparison of the relative stability of the *apo* and Ap4A-bound complexes. The results reveal that in the presence of Ap4A, the distance between AcuB_M188 and the Zn$^{2+}$ ion in the active site of AcuC is shorter than in the apo state (Fig. 5e, g, h), which correlates with the observed enhancement of Ap4A-mediated inhibition of AcuC (Fig. 4e). Likewise, variation of AcuB M188 to alanine diminished complex formation, supporting the important role of this residue for the AcuB-AcuC interaction (Supplementary Fig. 9).

To further characterize the conformational space explored during the simulations, we performed principal component analysis (PCA) on the concatenated trajectories after removing overall translational and rotational motions. A mass-weighted covariance matrix was constructed and diagonalized, and the first two principal components (PC1 and PC2), accounting for approximately 50% of the total variance, were used as reaction coordinates. Free energy landscapes were then derived through Boltzmann inversion of kernel density–estimated conformational probabilities (Supplementary Fig. 20). Both *apo* and Ap4A-bound systems exhibited two main basins, indicating the

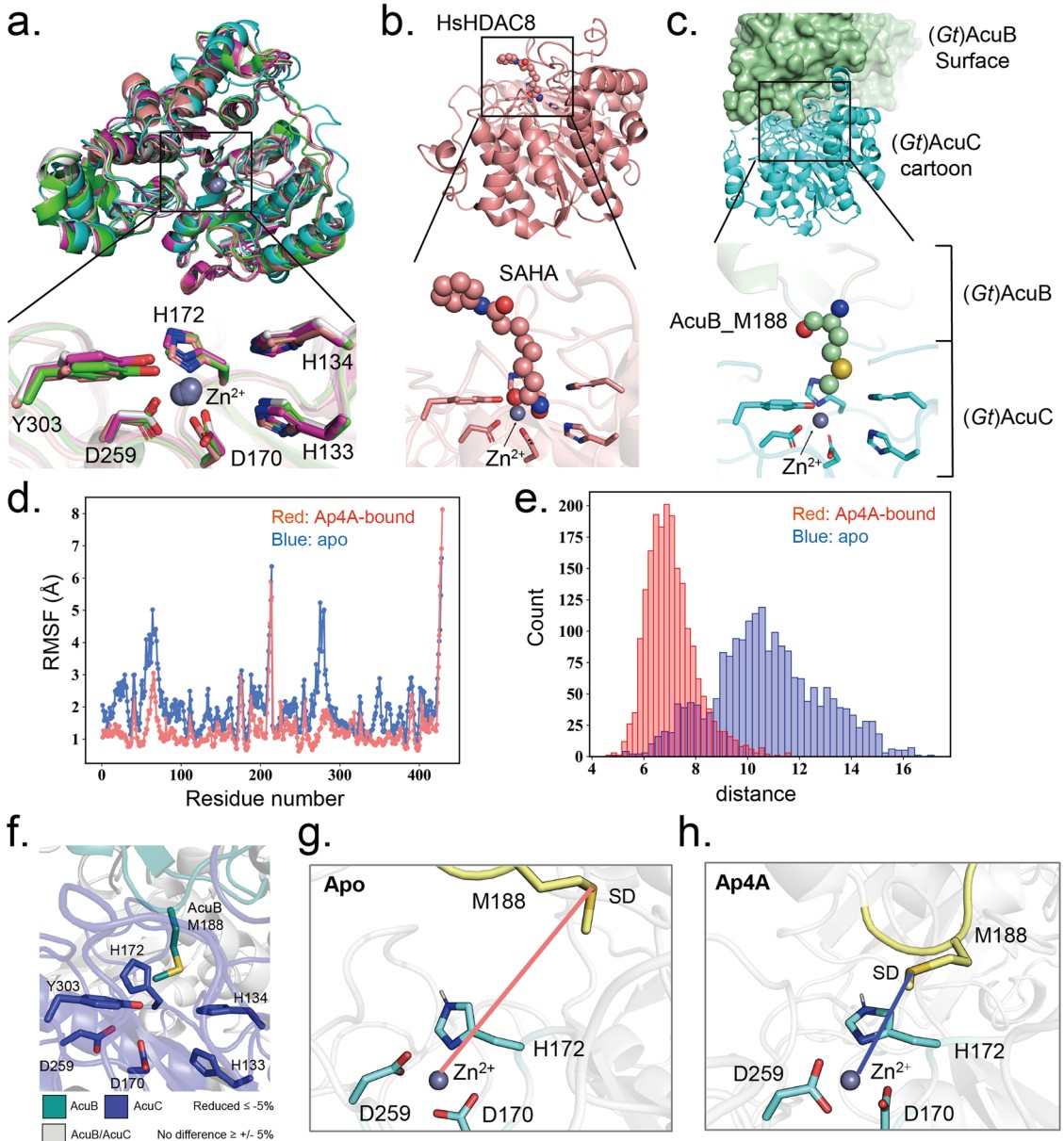

**Fig. 5 | Mechanism of AcuC inhibition by AcuB and its regulation *via* Ap4A.**
**a** Structural alignment of the overall structure and catalytic center of (*Gt*)AcuC (cyan, AlphaFold 3) with (*Hs*)HDAC1 (magenta, PDB ID: 4BKX), (*Hs*)HDAC2 (grey, PDB ID: 4LY1), (*Hs*)HDAC3 (green, PDB ID: 4A69), and (*Hs*)HDAC8 (orange, PDB ID: 1T69)[45–48]. **b** Coordination of suberoylanilide hydroxamic acid (SAHA; shown as spheres) to (*Hs*)HDAC8[41]. **c** Coordination of (*Gt*)AcuB (M188) within the (*Gt*)AcuC active center (AlphaFold 3) compared to (*Hs*)HDAC8 binding with SAHA. **d** Root Mean Square Fluctuation (RMSF) values (in Å) of Cα atoms obtained from molecular dynamics (MD) simulations in the absence of a ligand (apo; blue) and in the presence of Ap4A (red). **e** Distance (in Å) between the sulfur (SD) atom of AcuB_M188 and the zinc ion of AcuC in the active site during MD simulations, comparing the apo state (blue) and the Ap4A-bound state (red). **f** HDX analysis reveals structural differences in the (*Gt*)AcuC catalytic center when in complex, compared to its individual forms and (*Gt*)AcuB. **g–h** Close-up views of the active site in the *apo* (**g**) and Ap4A-bound (**h**) states. The carbon atoms of the AcuB_Zn²⁺-coordinating residues are shown in cyan, while AcuB_M188 and its associated loop are highlighted in yellow. Source data underlying (**d**, **e**) are provided as a Source Data file.

presence of two dominant conformational states. The relatively flat energy profiles suggest low energy barriers between these states, while subtle shifts in local minima imply that Ap4A binding may allosterically stabilize specific local conformations of AcuB that enhance its inhibitory interaction with AcuC. Together, these results suggest that Ap4A binding stabilizes the AcuB dimer and strengthens its association with AcuC through conformational restriction and local allosteric effects.

## Discussion
Here, we have demonstrated that AcuB proteins from two differently adapted species are Ap4A-regulated proteins and uncover a regulatory

network that might connect stress response, Ac-CoA homeostasis and protein acetylation.

## CBS domains as conserved binding partners of Ap4A
Our findings demonstrate that Ap4A binds to the CBS domains of AcuB, expanding its known repertoire of CBS domain interactions. Interestingly, Ap4A binds to AcuB in a manner which is different to previously reported binding partners[27,49,50]. In the pyrophosphatase (PPase) of *Clostridium perfringens*, a single Ap4A molecule binds diagonally across the CBS domains in a crisscross fashion[51]. In *B. subtilis* IMPDH (BsIMPDH) one Ap4A binds across two CBS domains in a horseshoe-like manner involving a magnesium ion[27]. In AcuB, two

Ap4A molecules bind in a parallel configuration, bridging the CBS domains. (Supplementary Fig. 21).

Moreover, our NanoDSF experiments also demonstrate that Ap4A increases the stability of AcuB. This finding suggests that this alarmone may act as a stabilizing agent upon heat stress – a condition known to induce Ap4A accumulation (around 60 μM) in vivo[27] – a role distinct from its previously reported functions as an inhibitor of enzymatic function[27,49–51].

## Relevance of Ap4A binding to AcuB

The *acuABC* operon encodes the acetyltransferase AcuA, the Ap4A-binding AcuB, and the deacetylase AcuC. Notably, the *acuB* and *acuC* genes overlap by four nucleotides at their stop and start codons, forming a 5′-ATGA-3′ junction-a signature feature of translationally coupled bacterial operons (Supplementary Fig. 22A). Such genetic coupling often facilitates the regulation of antagonistic protein-protein interactions.

Ribosome profiling (Ribo-seq) shows that AcuB is translated more abundantly than AcuC (Supplementary Fig. 22B). However, AcuB appears to be unstable (Fig. 2g) and might therefore only weakly inhibit AcuC in unstressed cells. Under normal growth conditions, deacetylase activity of AcuC is hence expected to predominate, ensuring proteome-wide deacetylation. In contrast, during stress, Ap4A binding stabilizes AcuB (Fig. 2g). This is likely to lead to a global increase in protein acetylation. Testing this hypothesis would require comprehensive acetylome profiling, although technical challenges have hindered such analyses so far.

This regulatory mechanism might well be overall conserved judged by the conservation of critical residues in in silico analyzed AcuB orthologs (Fig. 3e). Species specific adaptation of this process appears likely given that the effects of Ap4A on thermal stability in investigated AcuB orthologs is variable. Although we only observe a modest increase in stability for (Gt)AcuB in the presence of Ap4A, we

detect a significantly increased interaction of AcuB with AcuC specifically in response to Ap4A treatment (Fig. 4c).

## AcuC deacetylates a broad spectrum of substrates

AcsA[Ac] and TufA[Ac] were the only reported substrates for AcuC[13,32]. Given the significant structural differences and lysine residue variations between AcsA and TufA, it is plausible that AcuC targets additional proteins for deacetylation. This hypothesis is supported by in vivo data showing that deletion of *acuC* (together with *srtN*) leads to a 1.5-fold increase in the overall acetylation levels in the double mutant[52], and a reduction in sporulation of *B. subtilis*[53]. Our in vitro assays demonstrate that AcuC can deacetylate further proteins such as YabA (a negative regulator of DNA replication initiation)[34] and MurC (a key enzyme in cell wall biosynthesis) (Fig. 1). These findings are in good agreement with a global role for AcuC in controlling the acetylome (Fig. 6). Remarkably, AcuB inhibited the activity of AcuC for all tested targets. The presence of Ap4A stabilizes AcuB, the AcuB-AcuC complex and increases its inhibitory potential. Thus, we propose that Ap4A can impact multiple cellular processes via this mechanism, including metabolism, DNA replication initiation, and cell wall biosynthesis by keeping proteins in an acetylated state (Fig. 6).

## Synchronization of protein deacetylation and acetyl-CoA production

Acetyl-CoA synthetase (Acs) is a central metabolic enzyme ubiquitous across diverse organisms[54]. Its activity is regulated through reversible lysine acetylation at the C-terminal domain, a mechanism that is highly conserved[12,13,16]. Deacetylation of acetylated Acs (Acs[Ac]) not only reactivates the enzyme but also promotes acetyl-CoA production. Intriguingly, deacetylation events extend beyond Acs[Ac] to other proteins, such as histones, producing acetate that can be recycled by Acs, suggesting a synchronization of post-translational protein modifications with metabolite turnover[55]. Our findings add to this complexity by

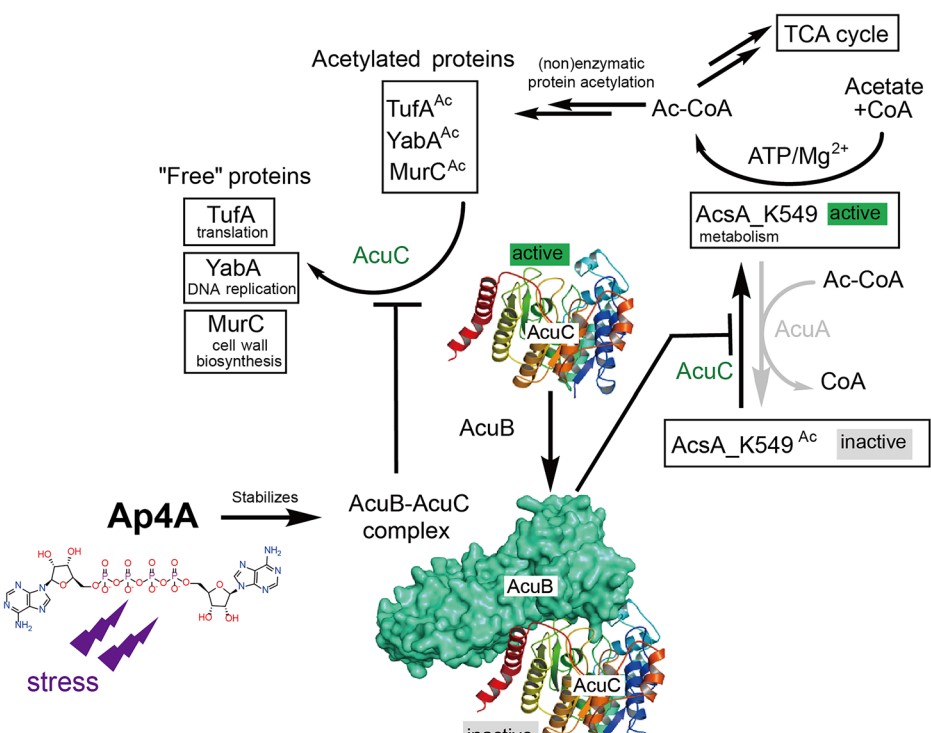

**Fig. 6 | Ap4A broadly regulates protein acetylation by stabilizing AcuB and enhancing the AcuB–AcuC complex.** Model illustrating the regulatory network characterized in this study. The Ap4A-AcuB-AcuC module has the capability to control the acetylome of *B. subtilis*. The alarmone Ap4A stabilizes AcuB and increase its inhibitory effect towards AcuC. AcuC is a regulator of protein acetylation and acetyl-CoA biosynthesis. This showcases a regulatory system coordinating stress response with protein acetylation and acetyl-CoA biosynthesis.

revealing that the deacetylase AcuC, inhibited by AcuB and Ap4A, plays a critical regulatory role in controlling the acetylation status of Acs together with many other *B. subtilis* proteins (Fig. 6). This interplay emphasizes a tightly coordinated mechanism that may enable cells to dynamically adapt their metabolism to environmental and physiological changes. Given the evolutionary conservation of AcuC and AcsA, it is possible that similar regulatory networks exist in eukaryotes, where they also may connect metabolic homeostasis to processes such as gene expression and stress response (Fig. 6).

### Inhibition of lysine deacetylase AcuC by the ACT domain of AcuB

Prior to our study, the ACT domain, commonly found in both eukaryotes and prokaryotes, was mainly recognized as a binding partner for small molecules[56,57]. We show here that such a domain can also bind in vicinity of active center of an enzyme to downregulate activity, which is a role of ACT domains. It is plausible that these domains fulfill similar functions in eukaryotes, including humans. Class I HDAC[2,44] are important drug targets for treatment of cancer and cardiovascular diseases[13,50,58,59] and deeper knowledge about the underlying regulatory mechanisms might be relevant to foster development of pharmaceuticals.

## Methods

### Protein Production and Purification

The genes encoding *B. subtilis* proteins (*Bs*)AcuB, (*Bs*)AcuC, and (*Bs*)AcsA were amplified from genomic DNA of strain NCIB 3610 (DK1042), while *(Gt)acuB* and *(Gt)acuC* genes were amplified from genomic DNA of *G. thermodenitrificans* NG80-2. Open reading frames were cloned into a pET24d-based plasmid using N- or C-terminal hexa-histidine tags via a Golden Gate cloning approach, utilizing BsaI restriction sites. For GST-tagged proteins, a similar cloning method was employed. The point mutants were generated by fusion PCR. Two overlapping PCR fragments containing the desired mutation were amplified using *(Bs or Gt)acuB* as the template and mutation-specific primers (Supplementary Tables 3 and 4). The fragments were subsequently fused in a second round of PCR to yield the full-length mutated gene. The resulting PCR product was cloned into the corresponding expression vector using a Golden Gate assembly approach as described above.

All proteins were overexpressed in *E. coli* BL21(DE3) (Novagen) in LB (Luria-Bertani) medium (tryptone: 10 g/L, yeast extract: 5 g/L, NaCl: 10 g/L), supplemented with 10 g/L D(+)-lactose monohydrate, 50 μg/mL kanamycin, or 100 μg/mL ampicillin (depending on the vector's resistance) for 16–20 h at 30 °C with vigorous shaking. After cultivation, cells were harvested by centrifugation at 4500 × g for 20 min at 4 °C. The cell pellet was then resuspended in buffer A (50 mM Tris-HCl pH 8.0, 150 mM NaCl, 20 mM MgCl$_2$, and 40 mM imidazole) and lysed using an LM10 microfluidizer (Microfluidics) at 18,000 psi. The lysate was subsequently centrifuged at 47,850 × g for 20 min at 4 °C.

The supernatant was applied to a HisTrap HP 1 mL column (Cytiva) equilibrated with buffer A. After washing with 10 mL of buffer A, proteins were eluted with 10 mL of buffer B (50 mM Tris-HCl pH 8.0, 150 mM NaCl, 20 mM MgCl$_2$, and 250 mM imidazole). The eluted proteins were concentrated to 2 mL using Amicon Ultracel-30K (Millipore) and subjected to size-exclusion chromatography (SEC) on a HiLoad 26/600 Superdex 200 pg column (Cytiva) equilibrated with 50 mM HCl pH 7.5, 150 mM NaCl, and 20 mM MgCl$_2$. Fractions containing the target proteins were further concentrated using Amicon Ultracel-30K (Millipore), snap-frozen in liquid nitrogen, and stored at −80 °C for subsequent analysis. Protein concentration was measured photometrically using a NanoDrop Lite (Thermo Fisher).

### Thermo stability assays

NanoDSF was used to measure the thermal stability of proteins using a Prometheus NT.48 instrument (NanoTemper Technologies, Germany). Protein samples were prepared at 1 mg/mL in SEC and with/without 500 μM Ap4A. Approximately 2 μL of each sample was loaded into UV-transparent capillaries (NanoTemper Technologies). The temperature was increased from 20 °C to 95 °C with a velocity of 1 °C per minute, while the intrinsic fluorescence at 330 nm and 350 nm was monitored. The unfolding transition temperature (Tm) was determined from the change in the fluorescence ratio (350/330 nm) using the ThermControl software (Nano Temper).

### In vitro pulldown assays

Strep-Tag pulldown assays were carried out using spin columns and filters from MobiTec. A 20 μL suspension of Strep-Tactin® coated magnetic beads (IBA) was placed in an eppi tube and resuspended in 450 μL of SEC buffer. After centrifuging at 4500 rpm for 1 minute, 200 μg of Strep-tagged protein was added to the beads and incubated on a rotating mixer for 15 min. The mixture was then centrifuged under the same conditions. The immobilized protein was washed with 500 μL of SEC buffer, followed by another centrifugation. Subsequently, 200 μg of the putative interaction partners were added in 400 μL of SEC buffer and incubated with the Strep-tagged protein on the rotating mixer for 30 minutes. After three washes with 500 μL of SEC buffer, the bound proteins were eluted using 50 μL of SEC buffer containing 50 mM D-Biotin (pH 7.5, Carbosynth). The samples were analyzed by SDS-PAGE stained with Coomassie blue. GST pulldown assays were performed similarly using MobiTec spin columns and filters. A 20 μL suspension of GST-Sepharose beads (GE Healthcare) was added to a spin column and resuspended in 450 μL of SEC buffer. After 1 minute of centrifugation at 4500 rpm, 200 μg of GST-tagged protein was added and incubated on a rotating mixer for 15 min. The mixture was centrifuged again, and the immobilized protein was washed with 500 μL of SEC buffer. Next, 200 μg of the interaction partner was added in 400 μL of SEC buffer and incubated with the GST-tagged protein on the rotating mixer for 30 min. Following three washes with 500 μL of SEC buffer, the bound proteins were eluted with 50 μL of SEC buffer containing 20 mM GSH (pH 8.0). The samples were separated by SDS-PAGE and stained with Coomassie blue for analysis.

### Isothermal titration calorimetry (ITC)

ITC experiments were conducted at a temperature of 25 °C using a MicroCal PEAQ-ITC instrument (Malvern Panalytical). Ligands and proteins were diluted in a buffer containing 20 mM HEPES-HCl, 20 mM MgCl$_2$, 20 mM KCl, 200 mM NaCl, pH 7.5. To measure the interaction between (*Bs*)AcuB or (*Gt*)AcuB and the different nucleotides, the purified protein was titrated in the sample cell up to final concentrations between 40 and 120 μM. The protein concentrations were predetermined by measuring absorbance at 280 nm. The ligand was placed in the titration syringe at a nominal concentration of 0.4–1 mM to saturate the protein sample during the titrations. Nucleotides were obtained from Jena Bioscience with a purity of greater than 95%. The assay consisted of 13 injections (first 0.4 μl, and then 3 μl per injection) with 150 seconds of spacing between each injection. The raw data were processed with the MicroCal PEAQ-ITC Analysis Software using the "one binding site" model and plotted using GraphPad Prism.

### Protein crystallization, X-ray diffraction, and structure determination

For Ap4A-bound (*Bs*)AcuB, a mixture of 300 μM (*Bs*)AcuB and 1 mM Ap4A was incubated for 10 min in ice. Initial crystallization trials were performed in SWISSCI MRC 2-well plates (Jena Bioscience) with a 30 μL reservoir, where 0.25 μL of protein was mixed with an equal volume of precipitant solution. Crystals appeared between two and seven days. To optimize crystallization conditions, hanging drop vapor diffusion was employed with a 1 mL reservoir. Drops were set up by mixing 1 μL of precipitant solution with varying volumes of protein solution (0.5 μL, 1 μL, and 2 μL). Ap4A-bound (*Bs*)AcuB crystals were successfully

grown in a solution containing 0.2 M sodium chloride, 0.1 M imidazole (pH 8.0), and 1.0 M potassium/sodium tartrate.

X-ray data were collected under cryogenic conditions at the ID23-1 beamline of the European Synchrotron Radiation Facility (ESRF) and at the P13 beamline of DESY. Data processing, integration, and scaling were performed with XDS and XSCALE[60]. The structures of Ap4A-bound (*Bs*)AcuB were solved by molecular replacement with PHASER[61], employing the crystal structure of a hypothetical protein, TTHA0829 from *Thermus thermophilus* HB8 (PDB-ID: 5AWE[40]). This hypothetical protein shared approximately 30% sequence identity (Supplementary Fig. 12). The structure was refined to $R_{Work}/R_{Free}$ values of 0.2384 and 0.2832, respectively (Supplementary Table 1). Model building was conducted in Coot[62], and refinement was performed with PHENIX[63]. Figures were generated with PyMOL[64].

### In vivo pulldown assay and identification of interaction partners
Overnight LB cultures were used to inoculate 20 mL of MC medium to an OD600 of 0.08, which was then grown to an OD600 of 1.2–1.4 at 37 °C. MC medium was prepared by combining 2 mL of MC-Buffer 10×, 0.67 mL of 1 M MgSO₄, and sterile double-distilled water up to 20 mL. The MC-Buffer 10× was made by dissolving K₂HPO₄·3H₂O (35.099 g), KH₂PO₄ (13.0975 g), glucose (50 g), 25 mL of 300 mM Na₃-citrate, 2.5 mL of 22 mg/mL ammonium ferric citrate, casein (2.5 g), and potassium glutamate (5 g) in 250 mL, followed by sterile filtration and storage in frozen aliquots. The cells were mixed with 0.5–1.5 µg of DNA and incubated for 2 h. After centrifugation at 5000 × g for 3 min, the pellet was plated on LB agar with antibiotics and incubated overnight at 37 °C. MC medium was prepared by combining MC-buffer with 1 M MgSO₄ and sterile water, while MC-buffer 10x was made by dissolving various components and sterilizing the solution. For expression testing with pLIKErep, overnight LB + MLS cultures with 1 µg/mL erythromycin were used to inoculate 10 mL LB + MLS, grown to OD600 0.4–0.5, and induced with 27 µg/mL bacitracin. The cultures were harvested, lysed in buffer, and analyzed by SDS-PAGE. Strep-Tactin® coated magnetic beads (IBA) were used to capture Strep-tagged AcuB, which was then eluted with 50 mM D-biotin. The eluate was analyzed by SDS-PAGE and mass spectrometry.

Trypsin (0.1 µg solubilized in 50 mM NH₄HCO₃) was added to each sample and samples were incubated at 37 °C over-night. Peptides were desalted and concentrated using Chromabond C18WP spin columns (Macherey-Nagel, Part No. 730522). Finally, peptides were dissolved in 25 µL of water with 5% acetonitrile and 0.1% formic acid. Peptide concentration was measured using a Nanodrop (Thermo Scientific) and samples were diluted accordingly to 100 ng of peptides per µL. The mass spectrometric analysis of the samples was performed using a timsTOF Pro mass spectrometer (Bruker Daltonic). A nanoElute HPLC system (Bruker Daltonics), equipped with an Aurora C18 RP column (25 cm × 75 µm ID) filled with 1.7 µm beads (IonOpticks, Australia) was connected online to the mass spectrometer. A portion of 2 µL of the peptide-solution was injected directly on the separation column. Sample Loading was performed at a constant pressure of 800 bar. Separation of the tryptic peptides was achieved at 60 °C column temperature with the following gradient of water/0.1% formic acid (solvent A) and acetonitrile/0.1% formic acid (solvent B) at a flow rate of 400 nL/min: Linear increase from 2% B to 17% B within 18 min, followed by a linear gradient to 25% B within 9 min and linear increase to 37% solvent B in additional 3 min. Finally, B was increased to 95% within 10 min and hold at 95% for additional 10 minutes. The built-in "DDA PASEF-standard_1.1sec_cycletime" method developed by Bruker Daltonics was used for mass spectrometric measurement. Protein identification was performed using Proteome Discoverer 2.4 (Thermo Fisher Scientific, Germany).

### DRaCALA screen
Two open reading frame proteome overexpression libraries, each consisting of 5139 ORFs from *B. anthracis* str. Ames, were used for the DRaCALA screen. The distinction between the two libraries was the protein expression vector and N-terminal protein tag (pVL791-AmpR with 10xHis tag; pVL847-GentR with 10xHis-MBP tag). The libraries had previously been constructed and overexpressed in *Escherichia coli* BL21 lacI�q, then lysed and prepared as described in ref.[39]. Screening for binding targets of Ap4A was conducted by differential radial capillary action of ligand assay (DRaCALA). 32P-labelled Ap4A was diluted to 15 µM total concentration in binding buffer (10 mM Tris-Cl pH 7.5, 100 mM NaCl, 5 mM MgCl₂), added to lysates at 1:1 volume, incubated for 10 min with shaking, and then spotted onto nitrocellulose paper. The spotted DRaCALA reactions were dried, then exposed to a phosphoscreen and imaged with Typhoon FLA9000 scanner.

### Hydrogen/deuterium exchange mass spectrometry (HDX-MS)
Samples of AcuB, AcuC and AcuB/AcuC complexes were purified by size-exclusion chromatography prior HDX-MS and used at 50 µM stock concentration. Where indicated, Ap4A was employed at 1 mM stock concentration.

Preparation of samples for HDX-MS was supported by a two-arm robotic autosampler (LEAP Technologies) essentially as previously described in ref.[65]. In brief, HDX reactions were initiated by 10-fold dilution of the proteins (50 µM) with buffer (20 mM Tris pH 7.4, 150 mM NaCl, 20 mM MgSO₄) prepared in D₂O and incubated for 10, 30, 100, 1000 or 10,000 s at 25 °C. The exchange was stopped by mixing with an equal volume of pre-dispensed quench buffer (400 mM KH₂PO₄/H₃PO₄, 2 M guanidine-HCl; pH 2.2) kept at 1 °C, and 100 µl of the resulting mixture injected into an ACQUITY UPLC M-Class System with HDX Technology[66]. Non-deuterated samples were generated by a similar procedure through 10-fold dilution with buffer prepared with H₂O. The injected HDX samples were washed out of the injection loop (50 µL) with water + 0.1% (v/v) formic acid at a flow rate of 100 µL/min and guided to a column containing immobilized porcine pepsin kept at 12 °C. The resulting peptic peptides were collected on a trap column (2 mm × 2 cm), that was filled with POROS 20 R2 material (Thermo Scientific) and kept at 0.5 °C. After three minutes, the trap column was placed in line with an ACQUITY UPLC BEH C18 1.7 µm 1.0 × 100 mm column (Waters) and the peptides eluted with a gradient of water + 0.1% (v/v) formic acid (eluent A) and acetonitrile + 0.1% (v/v) formic acid (eluent B) at 30 µL/min flow rate as follows: 0–7 min/95–65% A, 7–8 min/65–15% A, 8–10 min/15% A. Eluting peptides were guided to a Synapt G2-Si mass spectrometer (Waters) and ionized with by electrospray ionization (capillary temperature and spray voltage of 250 °C and 3.0 kV, respectively). Mass spectra were acquired with the software MassLynx MS version 4.1 (Waters) over a range of 50–2000 *m/z* in enhanced high-definition MS (HDMSᴱ)[67,68] or high-definition MS (HDMS) mode for non-deuterated and deuterated samples, respectively. Lock mass correction was conducted with [Glu1]-Fibrinopeptide B standard (Waters). During separation of the peptides on the ACQUITY UPLC BEH C18 column, the pepsin column was washed three times by injecting 80 µL of 0.5 M guanidine hydrochloride in 4% (v/v) acetonitrile. Blank runs (injection of double-distilled water instead of the sample) were performed between each sample. All measurements were carried out in triplicates. Peptides were identified and evaluated for their deuterium incorporation with the softwares ProteinLynx Global SERVER 3.0.1 (PLGS) and DynamX 3.0 (both Waters). Peptides were identified with PLGS from the non-deuterated samples acquired with HDMSᴱ employing low energy, elevated energy and intensity thresholds of 300, 100 and 1000 counts, respectively and matched using a database containing the amino acid sequences of AcuB, AcuC, porcine pepsin and their reversed sequences with search parameters as follows: Peptide tolerance = automatic; fragment tolerance = automatic; min fragment ion matches per peptide = 1; min fragment ion matches per protein = 7; min peptide matches per protein = 3; maximum hits to return = 20; maximum protein mass = 250,000; primary digest reagent = non-specific; missed cleavages = 0; false

discovery rate = 100. For quantification of deuterium incorporation with DynamX, peptides had to fulfil the following criteria: Identification in at least 6 of the 12 non-deuterated samples; the minimum intensity of 10,000 counts; maximum length of 40 amino acids; minimum number of products of two and 0.1 products per residue; maximum mass error of 25 ppm; retention time tolerance of 0.5 min. All spectra were manually inspected and omitted, if necessary, e.g., in case of low signal-to-noise ratio or the presence of overlapping peptides disallowing the correct assignment of the isotopic clusters. Residue-specific deuterium uptake from peptides identified in the HDX-MS experiments was calculated with the software DynamX 3.0 (Waters). In the case that any residue is covered by a single peptide, the residue-specific deuterium uptake is equal to that of the whole peptide. In the case of overlapping peptides for any given residue, the residue-specific deuterium uptake is determined by the shortest peptide covering that residue. Where multiple peptides are of the shortest length, the peptide with the residue closest to the peptide C-terminus is utilized. Raw data of deuterium uptake by the identified peptides and residue-specific HDX are provided in Supplementary Table 2 and Supplementary Dataset 3.

## Activity assay for Ac-CoA recovery with AcsA$^{Ac}$

The acetylated AcsA (AcsA_K549$^{Ac}$) was prepared as described in ref. 30. The AcsA activity assay was performed by preincubating 12 µg of (*Gt*)AcuC with 4 µg of (*Gt*)AcuB at room temperature for 10 minutes. To assess the effects of Ap4A, 400 µM of Ap4A was added to the (*Gt*)AcuC-(*Gt*)AcuB mixture. The reaction was initiated by adding AcsA_K549$^{Ac}$ to a 200 µL reaction mixture containing 0.5 mM CoA, 2 mM ATP, 0.1 mM DTT, and 5 mM acetate. The reaction proceeded at room temperature, and 20 µL samples were taken at intervals of 1, 3, 6, 10, 30, and 60 minutes. The reaction was quenched by adding 40 µL of acetonitrile, followed by centrifugation at 16,200 x g to remove any precipitates. The resulting supernatant was analyzed using High-Performance Liquid Chromatography (HPLC) on an Agilent 1200 series system equipped with a Metrosep A Supp 5–150/4.0 column (Metrohm). Isocratic elution was performed with 90 mM $(NH_4)_2CO_3$ (pH 9.25) at a flow rate of 0.6 mL/min for 20 minutes, with detection carried out at 260 nm.

## Mass photometry

Mass photometry (MP) measurements were performed using a TwoMP mass photometer (Refeyn Ltd., UK) and controlled with the AcquireMP v2023 R1.1 software (Refeyn Ltd., UK). Microscope coverslips (1.5 H, 24 × 60 mm, Carl Roth) and Culture Well Reusable Gaskets (CW-50R-1.0, 3 × 1 mm; Grace Bio-labs, USA) were cleaned by alternating rinses with isopropanol and ultrapure water (Milli-Q, Merck, Germany) three times, followed by air-drying under a compressed air stream. The cleaned coverslips and gaskets were assembled and mounted on the mass photometer using immersion oil (ImmersolTM 519 F; Carl Zeiss, Germany). For each MP measurement, a drop of 19 µL of SEC buffer was applied to one well of the gasket and focused using the "Droplet dilution" option. This was followed by adding 1 µL of pre-diluted protein or protein mixture (details below) to reach a final concentration of 25 nM in the drop. The mass photometer was calibrated using a custom molecular weight standard containing proteins with known sizes ranging from 86 kDa to 344 kDa. MP measurements were recorded for 60 seconds at a rate of 100 frames per second. Data analysis was conducted using DiscoverMP v2023 R.1.2 software (Refeyn Ltd., UK). Peaks in the data were manually selected and fitted with a Gaussian model. The total number of counts predicted by the Gaussian fit, along with percentage values relative to the total number of binding events, were extracted to quantify and compare the oligomeric states of the proteins under study.

## Western blot

The YabA$^{Ac}$, TufA$^{Ac}$, MurC$^{Ac}$ were acetylated by incubating the proteins with 1 − 5 mM Ac-CoA overnight. The reaction mixture was then diluted

and concentrated using Amicon® Ultra Centrifugal Filter concentrators with a 10 kDa or 30 kDa molecular weight cut-off until the Ac-CoA concentration was reduced to below 100 nM. For activity assays, 20 µg of (*Gt*)AcuC was pre-incubated with 5 µg of (*Gt*)AcuB for 10 minutes. To evaluate the effects of Ap4A, 400 µM of Ap4A was added to the (*Gt*)AcuC-(*Gt*)AcuB mixture. The acetylated proteins (10 µg) were then added to the enzyme mixture, and the reaction was carried out at 37 °C. Samples were collected after 1 and 2 hours. The samples were subjected to SDS-PAGE and transferred to a PVDF membrane using the BIO-RAD Trans-Blot Turbo system (7 minutes, 1.3 A, 25 V). The membrane was blocked for 1 hour at room temperature with 10% non-fat dry milk (NFDM) in 1x TBST (20 mM Tris-HCl, pH 7.5, 150 mM NaCl, 0.1% (v/v) Tween-20). Anti-acetyl-lysine antibodies (1:1500) were incubated overnight at a temperature of 4 °C, followed by three washes with 1x TBST. The membrane was then incubated with anti-rabbit IgG-alkaline phosphatase antibodies (1:1500) for 1 hour at 4 °C. Signals were detected using the ECL Prime system and visualized with a Fusion-SL chemiluminescent imager (Peqlab). Full scan blots are provided in the the Source Data file.

## Analytical size-exclusion chromatography (SEC)

Purified N-Strep-(*Gt*)AcuB-C-His and N-His-(*Gt*)AcuC were prepared at 100 µM in a buffer containing 50 mM Tris-HCl pH 7.5, 150 mM NaCl, 20 mM MgCl$_2$, and incubated for 10 minutes at room temperature. Subsequently, 100 µL of the mixture were injected at 10 °C onto a pre-equilibrated S75 300/10 GL analytical size-exclusion column (GE Healthcare, Munich, Germany) using an Äkta system (UNICORN 7.6; Cytiva). The data were plotted using GraphPad Prism (GraphPad Prism Corp., San Diego). Single proteins were also injected and processed using the same procedure.

**Accelerated MD Simulations of the AcuB Dimer at 300 K.** The three-dimensional structures of the AcuB dimer in both the apo and Ap4A-bound states were generated using SWISS-MODEL[69], with the crystal structure of the (*Bs*)AcuB–Ap4A complex resolved in this study serving as a template. The Ap4A ligand was docked into the active site of AcuB using LeDock[70] software. For molecular dynamics (MD) simulations, the AMBER ff14SB force field[71] was applied to the protein, while the General Amber Force Field (GAFF)[72] was used for the Ap4A ligand. The systems were solvated in an explicit TIP3P[73] water box with a minimum buffer distance of 16 Å, and sodium ions were added to neutralize the system. Energy minimization was performed in two stages: initial relaxation of solvent and ions, followed by minimization of the entire system. The system was gradually heated to 300 K over 50 picoseconds (ps) using the NVT ensemble, followed by 500 ps of equilibration under NPT conditions. Production simulations were conducted using AMBER22 for 200 ns under accelerated MD (aMD)[74] conditions to enhance sampling efficiency. Long-range electrostatics were treated using the Particle Mesh Ewald (PME) method[75], with a cutoff of 12 Å for nonbonded interactions. Covalent bonds involving hydrogen atoms were constrained using the SHAKE algorithm[76], and the time step was set to 2 femtoseconds (fs).

## MD Simulations of the AcuB$_2$-AcuC$_1$ Trimer at 450 K

To further investigate the interaction between AcuB and AcuC, a 2AcuB_1AcuC trimer model was generated using AlphaFold3[35]. The coordination environment of the Zn$^{2+}$ ion in the active site of AcuC was parameterized using the "MCPB.py" modeling tool of AmberTools22[77], ensuring proper metal-ligand interactions. The protonation states of all titratable residues were determined based on their p$K_a$ values, which were calculated using the PROPKA software[78]. The system was prepared using the AMBER ff14SB force field for proteins and GAFF for Ap4A, while partial atomic charges were obtained from the RESP[79] method using the B3LYP/6-31 G* level of theory. The system was solvated in a TIP3P water box with a 16 Å buffer, and neutralizing ions were added. Following the same two-stage minimization protocol, the

system was gradually heated to 450 K under NVT conditions, followed by 500 ps of NPT equilibration. A 200 ns production run was performed using AMBER22 with periodic boundary conditions. Electrostatic interactions were handled using the PME method, and a 12 Å cutoff was applied for van der Waals interactions. The SHAKE algorithm was used to constrain bonds involving hydrogen atoms, allowing for a 2 fs time step. Trajectory analysis was performed using Visual Molecular Dynamics (VMD)[80] software to evaluate the structural dynamics of the simulated systems. Root Mean Square Fluctuation (RMSF) values of Cα atoms were computed to assess conformational flexibility across the simulation. For molecular dynamics trajectories see Supplementary Dataset 4. Summary and checklist tables of MD simulations are shown as Supplementary tables 5 and 6, respectively. Evidence for system equilibration is shown in Supplementary Fig. 23.

## Statistics and Reproducibility
All experiments were performed at least twice with similar results if not noted otherwise, and one representative dataset is shown in Figs. 1c, 1e–1g, 2c, 3f, and 4b.

## Reporting summary
Further information on research design is available in the Nature Portfolio Reporting Summary linked to this article.

## Data availability
The atomic coordinates have been deposited in the Protein Data Bank (PDB) under accession code 9QS7 (the structure of AcuB bound to Ap4A). Other atomic coordinates relevant to this study can be accessed via the PDB under accession codes 4BKX (the structure of HDAC1); 4LY1 (the structure of Human HDAC2); 4A69 (the structure of HDAC3); 1T69 (the structure of human HDAC8 complexed with SAHA); 1ZZ1 (the structure of a HDAC-like protein from *Alcaligenaceae bacterium* with SAHA bound). The HDX-MS data have been deposited to the ProteomeXchange Consortium via the PRIDE[81] partner repository with the dataset identifier PXD064203. The source data underlying Figs. 1c – 1g, 2c, 2f, 2g, 4b–4d, 5d, 5e and Supplementary Figs. 1, 4 – 7, 9, 11, 16, and 22 are provided as a Source Data file. Source data are provided with this paper.

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

## Acknowledgements

We would like to thank the Max-Planck-Society for support (to G.B., L.Z., and G.K.A.H.). We acknowledge support by the German Research Council (DFG) through the core facility for HDX-MS (project 324652314 to G.B. and U.L.), and the Research Training Group 2937 "Nucleotide Metabolism in Microbes" (to G.B. and P.B.). We thank the LOEWE initiative for funding through the exploration grant "Ap4-All" (to G.B. and J.F.). G.B. is grateful to the European Research Council (ERC) for support through the project "KIWIsome" (Grant agreement number: 101019765). J.F. acknowledges funding from the DFG (grant ID FR-3586/2-1). This work was supported by the National Institutes of Health Grant 5R35GM127088 and a USDA Hatch Formula Award (to J.D.W.). M.K.M.Y. was funded, in part, by Molecular Biosciences Training Grant (MBTG) 5T32GM007215-43 from NIH. L.Z. would like to thank Pietro Giammarinaro and Yifei Du for support at the beginning of the project. We acknowledge Michael Lammers (University Greifswald, Germany) for fruitful discussions and sharing of data prior to publication.

## Author contributions

L.Z., J.F., J.D.W., and G.B. designed the research, supervised the project and wrote the paper with input from all authors. M.K.M.Y and B.X.L. performed the DRaCALA screen. W.S. conducted and analyzed the HDX experiments. Z.G. and A.L. did the MD simulations. E.J.-K. performed ITC measurements. L.Z., F.B., P.B., C.-N.M., and J.P.F. generated constructs, purified proteins, performed enzyme assays and did the structural characterization. U.L. performed the proteomics experiment and related analysis. L.Z., M.G., and G.H. performed and analyzed the Mass-Photometry experiments. A.T. analyzed the RiboSeq data. All authors contributed to data analysis.

## Funding

## Competing interests

The authors declare no competing interests.
