## [Transparent Peer Review file · Nature Communications]

A protein adaptor mediating Ap4A-dependent control of protein acetylation.

Corresponding Author: Professor Gert Bange

Version 0:

Reviewer comments:

Reviewer #1

(Remarks to the Author)

Reversible lysine acetylation is a highly conserved post-translational modification that regulates a wide range of cellular processes in organisms. In bacteria, acetylation level precisely controls critical processes such as metabolic adaptation and virulence, which holds significant implications for understanding the physiological functions of protein acetylation and designing antibacterial targets. This study primarily reveals the previously uncharacterized protein AcuB, which is stabilized by the alarmone diadenosine tetraphosphate (Ap4A), leading to the functional inhibition of the deacetylase AcuC. The physiological significance of AcuB is emphasized.

Through a series of experiments, including pulldown assays, protein purification, HDX-MS, isothermal titration calorimetry and so on, the interaction domains of AcuC-AcuB and Ap4A-AcuB complex were determined. Specifically, during stress conditions, two Ap4A molecules bind symmetrically to the cystathionine beta-synthase domain of AcuB, thereby enhancing the stability of AcuB and subsequently inhibiting the deacetylase activity of AcuC. This inhibition affects the deacetylation of multiple substrates, including acetyl-CoA synthetase and translation elongation factors.

Overall, this study is highly intriguing and innovative, with rigorous experimental design and reliable results. It unveils a regulatory network connecting stress response, protein acetylation, and acetyl-CoA biosynthesis, and innovatively identifies the physiological role of the previously uncharacterized protein AcuB. However, the study has the following main limitations:

Major comments:

1. It lacks a global acetylome analysis to validate the hypothesis that protein acetylation increases under stress conditions.
2. Proteomics techniques should be employed to systematically screen and identify potential substrate proteins of AcuC, further elucidating its function and related regulatory networks.
3. The study primarily focuses on *Bacillus* species; the universality of this mechanism in other organisms should be thoroughly explored. In *Geobacillus thermodenitrificans*, the addition of Ap4A results in a relatively modest increase in the thermal stability of AcuB. These results raise the possibility that this process is controlled by species-specific regulatory mechanisms.
4. The authors demonstrate that under stress conditions, Ap4A stabilizes AcuB and inhibits AcuC's deacetylase activity. Whether this mechanism can be applied to antibacterial drug design or used to optimize microbial metabolic efficiency for industrial production through engineered acetylation regulatory circuits remain to be explored.
5. The term "(Bs)AcuB_R33E" is mentioned for the first time in Supplementary Figure 8a, but its origin (e.g., mutation rationale, experimental design) has not been explained.

Minor comments:

1. The horizontal axis label is missing in Figure 3e.
2. The phrase "in the (Gt)AcuC_R33E mutant" should be corrected to "in the (Gt)AcuB_R33E mutant" on page 6, line 40.

Reviewer #2

(Remarks to the Author)

The manuscript "A protein adaptor mediating Ap4A-dependent control of protein acetylation" by Bange, Wang and collaborators, demonstrates the interaction both mesophilic and thermophilic homologs between AcuB and AcuC, and suggests the relevance of Ap4A as an alarmone able to modulate such binding and, as a consequence, downregulating the deacetylase activity of AcuC.

The article is well written and organized, providing a number of convincing experimental data that confirms the proposed model of interaction of Ap4A with CBS domains in AcuB, as well as the distinctive stability of the latter and its interaction with AcuC.

The main concern of this referee is in the use of too simplistic (despite fully standard) and importantly limited molecular dynamics simulations to explain the molecular mechanism of AcuB/AcuC binding, as well as the effect of the presence of Ap4A. Despite the simulation apparently confirming the experimental results, only partial information is provided with respect to the differential stability of the apo and Ap4A bound states, and they are mostly based in the distance between the sulfur atom in AcuB_M188 and the catalytic zinc ion in AcuC. It is not clear than a single 200ns trajectory can be determinant for the conclusions claimed in Fig 5e, 5g and 5h and further analysis, at least, of the simulation are needed. For example, how good are the docking calculations? what other conformations (if any) are explored during the MD run for each system? Giving the fact that two apparently distant conformations are obtained in the simulation, a proper analysis of the free energy landscape using tools like Markov models or alike would enhance the interest of the simulations in this article. It is obvious that such analysis would require larger sampling either by extending the simulation or by adding additional replicas. Otherwise, this section of the article does not seem to be justifiable.

Minor issues:

-in page 3, lines 10-19, it appears confusing information in the labelling of AcuC and AcuB, with (Gt) prefixes in some sentences and not present in others. It is not clear if this difference is intentional, reflecting generic vs specific concepts, or just a typo.

-page 3. line 20-22. The sentence seems to lack correct grammar.

-page 4. lines 26-40. A value of $K_d = 0.39$ micromolar is said to be a "strong binding affinity", when a most realistic description would be moderate binding affinity.

-page 5. line 2. The variant (Bs)AcuB_R33E is first mentioned and it was not mentioned "above".

-page 5. lines 9-10. I suggest using the sentence "Thus, AcuB is an Ap4A stabilized AcuC inhibitor"

Reviewer #3

(Remarks to the Author)

The paper by Zheng et al., addresses the mechanism of action of AcuB whose functions have remained unknown for decades. AcuB is a regulator of AcuC which deacetylates a broad spectrum of substrates. Based on this broad specificity, the authors proposed that AcuC has a global role in controlling the acetylome. The interaction of AcuB with AcuC is mediated by the second messenger alarmone A4pA. Binding of the alarmone stabilises AcuB and favours the interaction with AcuC. The authors proposed based on HDX-MS measurements and pulldown assays that while the CBS domain of AcuB is the one that interacts with A4pA, it is the ACT domain of AcuB that engages AcuC and blocks the active site of the enzyme.

This is a solid and well written manuscript that deconstructs different aspects of the inhibition of AcuC by AcuB and the regulation of AcuB by the A4pA alarmone.

General comments:

- While the interaction between Ap4A and AcuB is well supported by biochemical, biophysical and structural data the interaction between AcuB and AcuC relies only on HDX-MS data, therefore the predicted binding interface should be further supported by point mutations.

- It is undisputed that the protein binds preferentially Ap4A, however the affinity of AcuB for ADP is not negligible. In this context, the K_d differential between these molecules is less relevant, how do the authors rationalise this? Can the authors provide evidence from competition experiments?

- Is there a technical reason why ITC could not be used here? The reported methods are both qualitative and don't much mechanistic information that could explain the role of Ap4A on the stabilization of the AcuB-AcuC complex

- From the calorimetry experiments shown in Fig 3 : Can the authors clarify the molar ratio from the ITC measurement? From the figure it looks like a molar ratio of 0.5 Ap4A : 1 AcuB, suggesting that only one molecule of Ap4A is bound to the dimer. This seems to differ with the crystal structure which contains 2 Ap4A...

- In the Discussion section, perhaps should tone down the first section, the importance of the discovery is very clear, so there is no need to repeat this in three different ways. In addition, in my opinion the separation of the Discussion into small sections reduces somewhat the impact of the finding since all the results are only presented in a smaller context. I believe a more overarching text that combines all sections may address this issue.

Minor comments:

- Page 5, line 18: Perhaps the shape of the protein should be renamed for non Star Trek connoisseurs ...

Page 8 line 22: *Clostridium perfringens* should be in italic.

- The figures with the HDX differences plotted on the structures are not very clear, could the authors plot the HDX in surface representation as well? With the ribbons and the shadows is a rather difficult to analyse the figures.

Reviewer #4

(Remarks to the Author)

In this manuscript Zheng et al describes a novel regulatory pathway linking protein (de)acetylation and stress response via the alarmone Ap4A. Authors report interaction of previously uncharacterized protein AcuB with the AcuC deacetylase, which results in the inhibition of its activity. They demonstrate that otherwise unstable AcuB is stabilized by binding of alarmone Ap4A. They report the structure of AcuB bound to Ap4A, showing extensive interaction with CBS domain which results in stabilization of the homodimer and further impacts the stability of AcuB/AcuC complex. Moreover, this represents yet another unique mode of Ap4A binding, different from other (few) examples. Overall, this study is well designed and quite complete, it characterizes unprecedented function of AcuB and uncovers the pathway in which it is involved. It is novel and well written and will certainly be of great interest across the fields of post-translational modifications, stress response and metabolism.

The only weakness of this study is that different parts are performed using Acu proteins from two different bacteria. For example, the affinity of Ap4A for *Bacillus* AcuB is much bigger than that of *Geobacillus*, and the stabilizing effect is most likely much stronger. Hence, while Ap4A effect is much stronger in the *Bacillus* model, the studies on conformational changes and inhibitory function were performed in *Geobacillus* model. Maybe this is the reason, why the impact on interaction seen by pulldown (figure 4c, lane 3) is not very big as compared to condition without Ap4A. In the printed version of draft, it is barely visible. Nevertheless, this limitation is due to a technical constraint, since authors cannot work with AcuB from *Bacillus* which aggregates during purification. However, since the *Geobacillus* AcuB is already relatively stable, I am wondering if the interpretation of the Ap4A effect on its interaction with AcuC is comparable to the case of *Bacillus*. This issue could be better communicated in the text. Otherwise, I have only minor suggestions:

Page 2, Lines 16-19 – I missed the transition between these two paragraphs. Authors statement is very broad, i.e. that they set out to fill the gap of knowledge of Ap4A mechanisms across domains of life, but they actually detail one particular case. Although the argument is valid, it is also detached from what comes immediately after, since authors first talk about interactions between Acu proteins, and only later present role of Ap4A. I would consider smoother transition, maybe including explanation why authors selected Acu pathway for investigation. I suppose the inspiration came from DRaCALA experiments and presence of ACT domain in AcuB.

Page 2, Line 21 – abbreviation or description for ACT domain should be provided at first mention.

Figure 1 d - Authors should be more precise in the legend of what is represented in this panel. Typically the minus sign would mean that there is no enzyme, but here it means inactivated (acetylated AcsA_K549ac), which is also present in all the conditions, but further supplemented with Zn²⁺ and AcuC and AcuC/AcuB (also with Zn²⁺?). If authors indicated directly on the figure that AcsA_K549 was present in all conditions, the figure would be immediately easier to interpret.

Authors should provide SEC calibration curve (Mw standards) to appreciate the sizes/size differences of the peaks in Figure 1 c.

Supplementary figures 5 and 6 – the first conditions addressed in the text are the ones with AcuB/AcuC, and only later those with Ap4A. The HDX figures could follow (top to bottom) this outline as well.

Page 4, Line 15 – the comparison with the structure of homolog from *Alcaligenaceae* bacterium is very brief and its relevance is not very clear. Was it solved with ligand? If this comparison is useful to define the active site of (Gt)AcuC, authors should be clearer in the text and could also provide supplementary figure with structure alignment indicating the predicted active site, maybe also in Figure 2b. Currently, the active site is only discussed much later in the last part of results.

Page 5, line 2, the description of R33E mutant is not provided above, it is given below at the line 38.

Page 5, line 18 – not being a fan of Star Trek, I personally had to google the term to understand what is “starship enterprise”. As for TTHA0829 protein, it is worth mentioning that it is a protein of an unknown function whose structure was previously reported.

Figure 3 d – for clarity, it would be better that carbons of both of the Ap4A molecules would differ from carbons of AcuB. Now one of right Ap4A molecule has the same color as one of the AcuB's.

Page 6, line 4 – for consistency please use the same type of abbreviations for amino acids, either one letter code or three letter code.

Figure 4c, it would be easier to interpret the figure if y axis would be called more specifically, i.e. molar ratio of which proteins?

Figure 4d, same comment as for figure 1d – it would be easier if the presence of AcsA_K549ac would appear in the legend or axis name, for example “AcCoA (peak area) in presence of AcsA_K549ac”.

Page 7, line 27 – what is the reason to perform MD simulations at 300K and then have a drastic shift to 450K? Wouldn't proteins denature at this temperature? As for non-expert in MD simulations, it sounds odd to me.

Figure 5 d – for easier interpretation legend to what is red, or blue could be provided directly in the figure panel.

Figure 6 a seems to be slightly alien to the study, maybe figure 6 b is sufficient to make to point.

Page 8, line 29 – In this study, the kD of Ap4A interaction with AcuB was measured to be about 0.4 uM. Authors could put this measurement in the biological context here, i.e. is it known what concentration of Ap4A in the cell under steady state conditions or upon stress?

Figure 6 c – I am not sure what is the reason to represent the AcuC in ribbon, and AcuB in volume? Esthetically it would be better to use one type of representation. Ap4A molecule could also be shown. The scheme could present the stabilizing effect of Ap4A on AcuB more clearly.

Version 1:

Reviewer comments:

Reviewer #1

(Remarks to the Author)

The author's response is purely textual and does not include experimental data, such as a global acetylome analysis or proteomics screening, which are typically used to identify AcuC's substrate proteins. The omission of such data is likely due to technical challenges or other constraints.

Therefore, the response does not fully resolve or rebut the core concerns raised in my original query.

Reviewer #2

(Remarks to the Author)

The response from the authors and their improvements to the manuscript are acceptable for this referee

Reviewer #3

(Remarks to the Author)

As mentioned in my previous report, this is a very interesting and well constructed work.

The authors have addressed all my concerns and comments in their revision and I am satisfied with the final manuscript.

Reviewer #4

(Remarks to the Author)

In the revised version of the manuscript authors have improved clarity of the manuscript and strengthened some parts of the study with additional results, such as mutations at the protein interaction surface and additional MD simulations. Authors have fully responded to my requests and suggestions and to majority of other reviewers concerns. I have no further comments, and I believe that this manuscript will be valuable contribution to several fields of studies as I have mentioned in my initial review.

REVIEWER COMMENTS

Reviewer #1 (Remarks to the Author):

Reversible lysine acetylation is a highly conserved post-translational modification that regulates a wide range of cellular processes in organisms. In bacteria, acetylation level precisely controls critical processes such as metabolic adaptation and virulence, which holds significant implications for understanding the physiological functions of protein acetylation and designing antibacterial targets. This study primarily reveals the previously uncharacterized protein AcuB, which is stabilized by the alarmone diadenosine tetraphosphate (Ap4A), leading to the functional inhibition of the deacetylase AcuC. The physiological significance of AcuB is emphasized.

Through a series of experiments, including pulldown assays, protein purification, HDX-MS, isothermal titration calorimetry and so on, the interaction domains of AcuC-AcuB and Ap4A-AcuB complex were determined. Specifically, during stress conditions, two Ap4A molecules bind symmetrically to the cystathionine beta-synthase domain of AcuB, thereby enhancing the stability of AcuB and subsequently inhibiting the deacetylase activity of AcuC. This inhibition affects the deacetylation of multiple substrates, including acetyl-CoA synthetase and translation elongation factors.

Overall, this study is highly intriguing and innovative, with rigorous experimental design and reliable results. It unveils a regulatory network connecting stress response, protein acetylation, and acetyl-CoA biosynthesis, and innovatively identifies the physiological role of the previously uncharacterized protein AcuB. However, the study has the following main limitations:

Major comments:

1. It lacks a global acetylome analysis to validate the hypothesis that protein acetylation increases under stress conditions.

We appreciate this insightful suggestion. We have attempted to perform a global acetylome analysis under stress conditions. Unfortunately, due to the limited quality of the obtained data, we were not able to reach a conclusive result as also stated in the discussion section of the initial submission: “*This is likely to lead to a global increase in protein acetylation. Testing this hypothesis would require comprehensive acetylome profiling, although technical challenges have hindered such analyses so far*”. We have not included our preliminary findings in the current version of the manuscript.

2. Proteomics techniques should be employed to systematically screen and identify potential substrate proteins of AcuC, further elucidating its function and related regulatory networks.

We thank the reviewer for this valuable suggestion. We agree that proteomics approaches are a possible venue for a more systematic identification of AcuC substrates and for elucidation of the broader regulatory network underlying these pathways. However, in the current study, we focus primarily on the regulatory mechanism of AcuB toward AcuC. Our results provide an initial indication that AcuC can act on multiple targets, laying the groundwork for future comprehensive proteomic investigations, which we feel are outside the scope of the current more mechanistic manuscript. Please note that we have previously shown, that AcsA – a well-established substrate of AcuC is encoded in close proximity in the genome – however it does not pull-down AcuC¹. Thus, a tight physiological interaction may not be mandatory for the deacetylation to occur. We will probably need a workaround e.g. a catalytic inactive AcuC variant in future attempts to find more AcuC substrates *in vivo*.

3. The study primarily focuses on Bacillus species; the universality of this mechanism in other organisms should be thoroughly explored. In Geobacillus thermodenitrificans, the addition of Ap4A results in a relatively modest increase in the thermal stability of AcuB. These results raise the possibility that this process is controlled by species-specific regulatory mechanisms.

Our bioinformatics analysis shows that the *acuA*, *acuB*, *acuC* operon mostly occurs in Bacillota phylum. The key binding residues (R33 and H34) for Ap4A are conserved across species (more than 200 species have been investigated, **Fig. 3e**). In the two tested bacteria the function of AcuB and also its preferential binding to Ap4A are conserved features, although both *Bacillus* species are adapted to a very different life-styles. This makes us believe that the Ap4A dependent regulation mechanism is conserved across species in principle. Please note that we detect Ap4A dependent stabilization of the AcuA-AcuB complex for *G. thermodenitrificans* proteins, something which we cannot measure for the *B. subtilis* proteins due to the instability of (Bs)AcuB.

4. The authors demonstrate that under stress conditions, Ap4A stabilizes AcuB and inhibits AcuC's deacetylase activity. Whether this mechanism can be applied to antibacterial drug design or used to optimize microbial metabolic efficiency for industrial production through engineered acetylation regulatory circuits remain to be explored.

We thank the reviewer for this valuable comment. We agree with the reviewer at this point, but think that addressing this point is probably outside the scope of our study.

5. The term "(Bs)AcuB_R33E" is mentioned for the first time in Supplementary Figure 8a, but its origin (e.g., mutation rationale, experimental design) has not been explained.

We have added this part in the method section

Minor comments:

1. The horizontal axis label is missing in Figure 3e.

We have added axis label as reviewer suggested.

2. The phrase "in the (Gt)AcuC_R33E mutant" should be corrected to "in the (Gt)AcuB_R33E mutant" on page 6, line 40.

We add the axis label in Figure 3e and change (Gt)AcuC_R33E to (Gt)AcuB_R33E as well

Reviewer #2 (Remarks to the Author):

The manuscript "A protein adaptor mediating Ap4A-dependent control of protein acetylation" by Bange, Wang and collaborators, demonstrates the interaction both mesophilic and thermophilic homologs between AcuB and AcuC, and suggests the relevance of Ap4A as an alarmone able to modulate such binding and, as a consequence, downregulating the deacetylase activity of AcuC.

The article is well written and organized, providing a number of convincing experimental data that confirms the proposed model of interaction of Ap4A with CBS domains in AcuB, as well as the distinctive stability of the latter and its interaction with AcuC.

The main concern of this referee is in the use of too simplistic (despite fully standard) and importantly limited molecular dynamics simulations to explain the molecular mechanism of AcuB/AcuC binding, as well as the effect of the presence of Ap4A. Despite the simulation apparently confirming the experimental results, only partial information is provided with respect to the differential stability of the apo and Ap4A bound states, and they are mostly based in the distance between the sulfur atom in AcuB_M188 and the catalytic zinc ion in AcuC. It is not clear that a single 200ns trajectory can be determinant for the conclusions claimed in Fig 5e, 5g and 5h and further analysis, at least, of the simulation are needed. For example, how good are the docking calculations? what other conformations (if any) are explored during the MD run for each system? Giving the fact that two apparently distant conformations are obtained in the simulation, a proper analysis of the free energy landscape using tools like Markov models or alike would enhance the interest of the simulations in this article. It is obvious that such analysis would require larger sampling either by extending the simulation or by adding additional replicas. Otherwise, this section of the article does not seem to be justifiable.

We sincerely thank the reviewer for this insightful comment. To further validate the robustness of our MD-based conclusions, we performed three independent MD simulations for both the apo and Ap4A-bound 2AcuB_1AcuC complexes. The results consistently revealed that in the presence of Ap4A, the distance between the sulfur atom of AcuB_M188 and the catalytic Zn²⁺ ion in AcuC was markedly shorter than in the apo state (average distances of 7.13 ± 0.87 Å and 11.5 ± 2.56 Å, respectively). More importantly, mutation of AcuB_M188A significantly reduced the AcuB-AcuC complex formation. These consistent findings strongly support the robustness of the conclusions drawn in the original manuscript (Figs. 5e, 5g, and 5h).

For the conformational analysis, each MD trajectory was preprocessed by removing overall translational and rotational motions. A mass-weighted covariance matrix was then constructed and diagonalized to obtain the principal components (PCs). The first two PCs, accounting for 50% of the total variance, were used as reaction coordinates. Using kernel density estimation, conformational probability distributions were computed, which were then converted to free energy landscapes via Boltzmann inversion. The resulting landscapes (Supplementary Fig. 19) provide a clear visualization of the conformational energy distributions for both apo and Ap4A-bound systems.

The free energy surfaces of both systems exhibit two shallow basins, indicating that the overall motion patterns are similar. However, subtle differences in local conformations suggest that the presence of Ap4A may allosterically stabilize regions critical for AcuC inhibition, consistent with our biochemical data.

Together, these results demonstrate that Ap4A binding stabilizes the AcuB dimer and strengthens its association with AcuC through conformational restriction and local allosteric effects. Although the current simulations were limited to hundreds of nanoseconds, the convergence of replicated trajectories and consistent distance and PCA analyses support the robustness of these conclusions. In future studies, extended simulations or enhanced sampling methods, such as Markov state modeling, could be employed to further resolve the detailed kinetic pathways underlying Ap4A-mediated regulation of AcuC activity.

Therefore, we added some more details to the revised manuscript:

“To further characterize the conformational space explored during the simulations, we performed principal component analysis (PCA) on the concatenated trajectories after removing overall translational and rotational motions. A mass-weighted covariance matrix was constructed and diagonalized, and the first two principal components (PC1 and PC2), accounting for approximately 50% of the total variance, were used as reaction coordinates. Free energy landscapes were then derived through Boltzmann inversion of kernel density–estimated conformational probabilities (Supplementary Fig. 19). Both apo and Ap4A-bound systems exhibited two main basins, indicating the presence of two dominant conformational states. The relatively flat energy profiles suggest low energy barriers between these states, while subtle shifts in local minima imply that Ap4A binding may allosterically stabilize specific local conformations of AcuB that enhance its inhibitory interaction with AcuC.....”

Minor issues :

-in page 3, lines 10-19, it appears confusing information in the labelling of AcuC and AcuB, with (Gt) prefixes in some sentences and not present in others. It is not clear if this difference is intentional, reflecting generic vs specific concepts, or just a typo.

We have added the relevant species

-page 3. line 20-22. The sentence seems to lack correct grammar.

We have made the change

-page 4. lines 26-40. A value of $K_d = 0.39$ micromolar is said to be a "strong binding affinity", when a most realistic description would be moderate binding affinity.

We have made the change accordingly

-page 5. line 2. The variant (Bs)AcuB_R33E is first mentioned and it was not mentioned "above".

We have made the change

-page 5. lines 9-10. I suggest using the sentence "Thus, AcuB is an Ap4A stabilized AcuC inhibitor"

We have made the change

Reviewer #3 (Remarks to the Author):

The paper by Zheng et al., addresses the mechanism of action of AcuB whose functions have remained unknown for decades. AcuB is a regulator of AcuC which deacetylates a broad spectrum of substrates. Based on this broad specificity, the authors proposed that AcuC has a global role in controlling the acetylome. The interaction of AcuB with AcuC is mediated by the second messenger alarmone A4pA. Binding of the alarmone stabilises AcuB and favours the interaction with AcuC. The authors proposed based on HDX-MS measurements and pulldown assays that while the CBS domain of AcuB is the one that interacts with A4pA, it is the ACT domain of AcuB that engages AcuC and blocks the active site of the enzyme.

This is a solid and well written manuscript that deconstructs different aspects of the inhibition of AcuC by AcuB and the regulation of AcuB by the A4pA alarmone.

General comments:

- While the interaction between Ap4A and AcuB is well supported by biochemical, biophysic and structural data the interaction between AcuB and AcuC relies only on HDX-MS data, therefore the predicted binding interface should be further supported by point mutations.

We thank the reviewer for raising this important point. We analyzed the interface of the (Gt)AcuB–AcuC complex supported by HDX-MS data, which revealed several interfacial interactions, including salt bridges between (Gt)AcuB_E4 and (Gt)AcuC_R362, (Gt)AcuB_D89 and (Gt)AcuC_K378, and (Gt)AcuB_E92 and (Gt)AcuC_R197, as well as π – π and hydrophobic interactions between (Gt)AcuB_W206 and (Gt)AcuC_Y197, and between (Gt)AcuB_M188 and (Gt)AcuC_F200/F143 (Supplementary Figure 7). Based on these analyses, we constructed several point mutants, including (Gt)AcuB_D89K, E92R, and M188A, as well as (Gt)AcuC_R197E, Y198A, and R362E. Subsequently, pulldown assays were performed to validate the predicted interactions. Among the AcuB mutants, D89K, E92R, and M188A showed markedly reduced complex formation; R197E and Y198A caused a moderate decrease; while R362E had little to no effect. These results are consistent with the predicted (Gt)AcuB–AcuC interface (**Supplementary Figure 7**).

The corresponding data have been added as **Supplementary Figure 7**, and the relevant description has been incorporated into the main text as follows: “More detailed analysis revealed that the interface between (Gt)AcuB and (Gt)AcuC is stabilized by multiple interactions, including salt bridges between (Gt)AcuB_E4–(Gt)AcuC_R362, (Gt)AcuB_D89–(Gt)AcuC_K378, and (Gt)AcuB_E92–(Gt)AcuC_R197, as well as π – π and hydrophobic interactions between (Gt)AcuB_W206–(Gt)AcuC_Y197 and (Gt)AcuB_M188–(Gt)AcuC_F200/F143 (Supplementary Fig. 9A). To validate these predicted interfacial contacts, we constructed single-point mutants (Gt)AcuB_D89K, E92R, M188A and (Gt)AcuC_R197E, Y198A, R362E, and performed pulldown assays (**Supplementary Fig. 9B**). The mutations D89K, E92R, and M188A in AcuB significantly reduced complex formation, while R197E and Y198A moderately weakened the interaction. In contrast, R362E showed minimal effect. These results confirm the key interfacial residues mediating the (Gt)AcuB–AcuC interaction.

- It is undisputed that the protein binds preferentially Ap4A, however the affinity of AcuB for ADP is not negligible. In this context, the Kd differential between these molecules is less relevant, how do the authors rationalise this? Can the authors provide evidence from competition experiments?

We have conducted competition assays using a ThermoShift assay to address the concern regarding the relative affinity of AcuB for ADP and Ap4A. The data clearly demonstrate that even when ADP is present at concentrations 5 times higher than Ap4A, the thermal shift induced by the binding of (Bs)AcuB remains comparable to that observed with Ap4A alone. This suggests that Ap4A preferentially binds to (Bs)AcuB, even in the presence of excess ADP. The fact that the thermal shift response is similar indicates that Ap4A is the dominant ligand, reinforcing its higher affinity for (Bs)AcuB. These results support the notion that, despite the presence of ADP, the binding of Ap4A is more energetically favorable and thus drives the interaction. Nevertheless, our data does not exclude a function for ADP binding in the absence of stress/Ap4A.

We have added following description in the main text: “It is worth mention that ADP also shows a μ M binding affinity to (Bs)AcuB. To further investigate the binding specificity of (Bs)AcuB to Ap4A, we performed competition experiments using a thermoshift assay. In these experiments, AcuB was incubated with both ADP and Ap4A, with ADP concentrations set 5 times higher than Ap4A. Despite the excess ADP, the thermal shift profile was nearly identical to that observed when Ap4A was present alone, confirming that Ap4A preferentially binds to (Bs)AcuB, even in competitive binding conditions. As ADP binds to (Bs)AcuB and changes its thermal stability – albeit to a lesser extent – it is possible that variations of ADP levels regulate AcuB activity in non-stressed cells, a phenomenon which merits future investigation ”

- Is there a technical reason why ITC could not be used here? The reported methods are both qualitative and don't much mechanistic information that could explain the role of Ap4A on the stabilization of the AcuB-AcuC complex

We also attempted to use ITC to investigate the role of Ap4A in stabilizing the AcuB–AcuC complex. However, under the conditions tested, we were unable to obtain a reliable Kd for the AcuB–AcuC interaction (**Fig. R1**). Unfortunately, ITC was not suitable to characterize this interaction, which is why we relied on complementary more qualitative methods such as pulldown and MP to evaluate the effect of Ap4A on complex stabilization.

Figure R1. ITC analysis of (Gt)AcuB–AcuC interaction measured at different protein concentrations.

- From the calorimetry experiments shown in Fig 3 : Can the authors clarify the molar ratio from the ITC measurement? From the figure it looks like a molar ratio of 0.5 Ap4A : 1 AcuB, suggesting that only one molecule of Ap4A is bound to the dimer. This seems to differ with the crystal structure which contains 2 Ap4A.

We appreciate this point. We noted that both (Gt)AcuB and (Bs)AcuB are not completely stable in solution and tend to precipitate at room temperature – even within 20 minutes, which might have affected the observed molar ratio in the ITC experiments, resulting in an apparent ratio of ~0.5 Ap4A per AcuB dimer. Given our structural analysis we think this is the most likely explanation

- In the Discussion section, perhaps should tone down the first section, the importance of the discovery is very clear, so there is no need to repeat this in three different ways. In addition, in my opinion the separation of the Discussion into small sections reduces somewhat the impact of the finding since all the results are only presented in a smaller context. I believe a more overarching text that combines all sections may address this issue.

We thank the reviewer for this suggestion and revised the first paragraph. However, we kept the different paragraphs as we believe it is helpful to discuss the different aspects of our finding.

Minor comments:

- Page 5, line 18: Perhaps the shape of the protein should be renamed for non Star Trek connoisseurs

We change the description, as following:

“(Bs)AcuB forms a homodimer, with the two chains cooperating in a configuration similar to that of its homolog TTHA0829 from *Thermus thermophilus* (PDB ID: 5AWE) (Fig. 3b, Supplementary Fig. 9).”

Page 8 line 22: *Clostridium perfringens* should be in *Italic*.

We have made the change

- The figures with the HDX differences plotted on the structures are not very clear, could the authors plot the HDX in surface representation as well? With the ribbons and the shadows is a rather difficult to analyse the figures.

We changed the representation of HDX projected onto structures for enhanced clarity. The shadows are omitted as suggested and alpha helices are now shown as cylinders. We decided against a surface representation as this obstructed the view onto critical parts, i.e., the Ap4A binding cleft of AcuB and the active site of AcuC.

Reviewer #4 (Remarks to the Author):

In this manuscript Zheng et al describes a novel regulatory pathway linking protein (de)acetylation and stress response via the alarmone Ap4A. Authors report interaction of previously uncharacterized protein AcuB with the AcuC deacetylase, which results in the inhibition of its activity. They demonstrate that otherwise unstable AcuB is stabilized by binding of alarmone Ap4A. They report the structure of AcuB bound to Ap4A, showing extensive interaction with CBS domain which results in stabilization of the homodimer and further impacts the stability of AcuB/AcuC complex. Moreover, this represents yet another unique mode of Ap4A binding, different from other (few) examples. Overall, this study is well designed and quite complete, it characterizes unprecedented function of AcuB and uncovers the pathway in which it is involved. It is novel and well written and will certainly be of great interest across the fields of post-translational modifications, stress response and metabolism.

The only weakness of this study is that different parts are performed using Acu proteins from two different bacteria. For example, the affinity of Ap4A for Bacillus AcuB is much bigger than that of Geobacillus, and the stabilizing effect is most likely much stronger. Hence, while Ap4A effect is much stronger in the Bacillus model, the studies on conformational changes and inhibitory function were performed in Geobacillus model. Maybe this is the reason, why the impact on interaction seen by pulldown (figure 4c, lane 3) is not very big as compared to condition without Ap4A. In the printed version of draft, it is barely visible. Nevertheless, this limitation is due to a technical constraint, since authors cannot work with AcuB from Bacillus which aggregates during purification. However, since the Geobacillus AcuB is already relatively stable, I am wondering if the interpretation of the Ap4A effect on its interaction with AcuC is comparable to the case of Bacillus. This issue could be better communicated in the text. Otherwise, I have only minor suggestions:

We thank the reviewer for the overall positive comments. We have discussed potential adaptations in both organisms in the revised paper. However, we are confident that key features are conserved. Both AcuB variants bind Ap4A and AcuC. While we indeed see stronger stabilization of (Bs)AcuB through Ap4A, we still observe increased interaction for (Gt)AcuB with AcuC upon Ap4A treatment. Both stabilization of AcuB and interaction AcuC might foster inhibition of AcuC *in vivo*, thus it appears to be a conserved regulatory scheme at least in principle.

We added the following paragraph to the discussion section:

Species specific adaptation of this process appears likely given that the effects of Ap4A on thermal stability in investigated AcuB orthologs is variable. Although we only observe a modest increase in stability for (Gt)AcuB in the presence of Ap4A, we detect a significantly increased interaction of AcuB with AcuC specifically in response to Ap4A treatment (Fig. 4c).

Page 2, Lines 16-19 – I missed the transition between these two paragraphs. Authors statement is very broad, i.e. that they set out to fill the gap of knowledge of Ap4A mechanisms across domains of life, but they actually detail one particular case. Although the argument is valid, it is also detached from what comes immediately after, since authors first talk about interactions between Acu proteins, and only later present role of Ap4A. I would consider smoother transition, maybe including explanation why authors selected Acu pathway for investigation. I suppose the inspiration came from DRaCALA experiments and presence of ACT domain in AcuB.

We thank the reviewer for pointing this out. We agree that the transition between the two paragraphs was abrupt and have now improved the flow of the introduction. Specifically, we have added a clearer rationale for selecting the acu operon for investigation which resulted from the CBS domain in AcuB, which we suspected to be an Ap4A binder also confirmed by our DRaCALA screening. The revised text now provides a smoother transition.

Page 2, Line 21 – abbreviation or description for ACT domain should be provided at first mention.

We have made the change

Figure 1 d - Authors should be more precise in the legend of what is represented in this panel. Typically the minus sign would mean that there is no enzyme, but here it means inactivated (acetylated AcsA_K549ac), which is also present in all the conditions, but further supplemented with Zn²⁺ and AcuC and AcuC/AcuB (also with Zn²⁺?). If authors indicated directly on the figure that AcsA_K549 was present in all conditions, the figure would be immediately easier to interpret.

We thank the reviewer for this valuable suggestion. We agree that indicating the presence of AcsA_K549^{Ac} in all conditions will improve clarity. We have now revised the figure to explicitly include AcsA_K549^{Ac} in each condition. In addition, we confirm that Zn²⁺ was supplemented in the AcuC reaction, and have updated the figure and legend accordingly.

Authors should provide SEC calibration curve (Mw standards) to appreciate the sizes/size differences of the peaks in Figure 1 c.

We have now included the molecular weight standards in Figure 1c to better illustrate the peak size differences and have added the corresponding details to the figure legend. To simplify the Figure, we provide the standard curve as **Supplementary Fig 4**.

Supplementary figures 5 and 6 – the first conditions addressed in the text are the ones with AcuB/AcuC, and only later those with Ap4A. The HDX figures could follow (top to bottom) this outline as well.

We amended the order in Supplementary Figs 5 and 6 as suggested.

Page 4, Line 15 – the comparison with the structure of homolog from Alcaligenaceae bacterium is very brief and its relevance is not very clear. Was it solved with ligand? If this comparison is useful to define the active site of (Gt)AcuC, authors should be clearer in the text and could also provide supplementary figure with structure alignment indicating the predicted active site, maybe also in Figure 2b. Currently, the active site is only discussed much later in the last part of results.

We thank the reviewer for this insightful comment. We have expanded the description of the structural comparison with the homolog from Alcaligenaceae bacterium (PDB ID: 1ZZ1) to better explain its relevance in defining the putative active site of (Gt)AcuC. In addition, we have added a structural alignment of (Gt)AcuC with 1ZZ1 as a new supplementary figure (**Supplementary Fig. 8**), and we now reference this comparison more clearly in the main text, including in the context of Figure 2b and the active-site discussion.

Page 5, line 2, the description of R33E mutant is not provided above, it is given below at the line 38.

We have supplied the information in the method section.

Page 5, line 18 – not being a fan of Star Trek, I personally had to google the term to understand what is “starship enterprise”. As for TTHA0829 protein, it is worth mentioning that it is a protein of an unknown function whose structure was previously reported.

We have changed the description as suggested.

Figure 3 d – for clarity, it would be better that carbons of both of the Ap4A molecules would differ from carbons of AcuB. Now one of right Ap4A molecule has the same color as one of the AcuB's.

We have changed the color as suggested.

Page 6, line 4 – for consistency please use the same type of abbreviations for amino acids, either one letter code or three letter code.

We have changed all with one letter code

Figure 4c, it would be easier to interpret the figure if y axis would be called more specifically, i.e. molar ratio of which proteins?

We have specified y axis in the Figure 4c legend, as following:

“The y axis molar ratio represents the N counts (peak of (Gt)AcuB-AcuC complex)/N counts (total) in percent”

Figure 4d, same comment as for figure 1d – it would be easier if the presence of AcsA_K549ac would appear in the legend or axis name, for example “AcCoA (peak area) in presence of AcsA_K549ac”.

We have added the details in legend as following: The peak area (Ac-CoA) is produced in the presence of AcsA_K549^{Ac}

Page 7, line 27 – what is the reason to perform MD simulations at 300K and then have a drastic shift to 450K? Wouldn't proteins denature at this temperature? As for non-expert in MD simulations, it sounds odd to me.

We thank the reviewer for this valuable comment. Indeed, proteins would denature at 450 K under experimental conditions. However, in MD simulations, transient heating is often employed not to reproduce physiological conditions but to accelerate conformational sampling and help the system overcome local energy barriers within a limited simulation time.

In our study, simulations at 300 K were used to evaluate the conformational stability of the AcuB dimer under near-physiological conditions, whereas simulations at 450 K were designed to enhance sampling efficiency and probe the relative stability and interaction tendencies of the AcuB–AcuC complex. Similar high-temperature MD protocols have been widely applied in previous studies to investigate protein–protein interaction stability, ligand dissociation, and

transition pathways²⁻⁴. Importantly, our conclusions are based on *comparative trends* between the apo and Ap4A-bound systems rather than on the absolute structural integrity of proteins at 450 K.

We have added following explanation in the main text: “It should be noted that the simulations at 450 K were not intended to mimic experimental conditions but rather to accelerate conformational sampling and enable comparison of the relative stability of the apo and Ap4A-bound complexes.”

Figure 5 d – for easier interpretation legend to what is red, or blue could be provided directly in the figure panel.

The color have now been added directly to Figures 5d and 5e

Figure 6 a seems to be slightly alien to the study, maybe figure 6 b is sufficient to make to point.

We have moved both Fig 6a and 6b to supplementary Figure as **Supplementary Fig 19**.

Page 8, line 29 – In this study, the K_D of Ap4A interaction with AcuB was measured to be about 0.4 μ M. Authors could put this measurement in the biological context here, i.e. is it known what concentration of Ap4A in the cell under steady state conditions or upon stress?

We have now included additional context regarding cellular concentrations of Ap4A under both steady-state and stress conditions (around 60 μ M). This information has been added to the text

Figure 6 c – I am not sure what is the reason to represent the AcuC in ribbon, and AcuB in volume? Esthetically it would be better to use one type of representation. Ap4A molecule could also be shown. The scheme could present the stabilizing effect of Ap4A on AcuB more clearly.

We thank the reviewer for this helpful suggestion. We agree that the stabilizing effect of Ap4A on AcuB should be presented more clearly and have revised Figure 6c accordingly, now also including the Ap4A molecule in the visualization.

Regarding the representation style, our intention in using a surface (volume) view for AcuB and a ribbon view for AcuC was to better illustrate the specific region of AcuB that inserts into the active site of AcuC, which we feel is more easily appreciated in this mixed representation. We hope the reviewer will agree that this choice improves the clarity of the interaction interface.

Reference

- 1 Zheng, L. *et al.* Regulation of acetyl-CoA biosynthesis via an intertwined acetyl-CoA synthetase/acetyltransferase complex. *Nature Communications* **16**, 2557 (2025).
- 2 Chen, J.-N., Dai, B. & Wu, Y.-D. Probability Density Reweighting of High-Temperature Molecular Dynamics. *J. Chem. Theory Comput.* **20**, 4977-4985 (2024).
- 3 Dai, B., Chen, J.-N., Zeng, Q., Geng, H. & Wu, Y.-D. Accurate Structure Prediction for Cyclic Peptides Containing Proline Residues with High-Temperature Molecular Dynamics. *The Journal of Physical Chemistry B* **128**, 7322-7331 (2024).
- 4 Abrams, C. F. & Vanden-Eijnden, E. Large-scale conformational sampling of proteins using temperature-accelerated molecular dynamics. *Biophys. J.* **98**, 26a (2010).

Point-by-point response letter

Reviewer #1 (Remarks to the Author):

The author's response is purely textual and does not include experimental data, such as a global acetylome analysis or proteomics screening, which are typically used to identify AcuC's substrate proteins. The omission of such data is likely due to technical challenges or other constraints. Therefore, the response does not fully resolve or rebut the core concerns raised in my original query.

We thank the reviewer for their valuable comments. We agree that global acetylome analysis or proteomics-based screening would be highly informative and represents an important direction for future follow-up studies.

Reviewer #2 (Remarks to the Author):

The response from the authors and their improvements to the manuscript are acceptable for this referee

We thank the reviewer for great comments and are pleased that the reviewer agree to publish our paper

Reviewer #3 (Remarks to the Author):

As mentioned in my previous report, this is a very interesting and well constructed work. The authors have addressed all my concerns and comments in their revision and I am satisfied with the final manuscript.

We thank the reviewer for excellent comments and are pleased that all concerns have been addressed.

Reviewer #4 (Remarks to the Author):

In the revised version of the manuscript authors have improved clarity of the manuscript and strengthened some parts of the study with additional results, such as mutations at the protein interaction surface and additional MD simulations. Authors have fully responded to my requests and suggestions and to majority of other reviewers concerns. I have no further comments, and I believe that this manuscript will be valuable contribution to several fields of studies as I have mentioned in my initial review.

We thank the reviewer for their constructive feedback and positive evaluation of the revised manuscript.